# Adenovirus maturation establishes the transcription competent packaging of its genome

Conradin Baumgartl[1,2,3], Simon Holzinger [2], Uwe Schwartz [2], Linda E Franken [4,7],
Kay Grünewald [4,5], Harald Wodrich [6✉] & Gernot Längst [2✉]

## Abstract

**Adenoviruses are human pathogens that more recently have gathered interest as tools for human gene therapy and vaccination. The maturation of the viral genome with associated proteins (core) remains largely unexplored. Here, we show that adenovirus core maturation is guided by features embedded in the viral DNA sequence, which primes the genome for transcription. Using DMS-seq to compare the accessibility of the nucleoprotein core structure before and after maturation (using the maturation deficient ts1 mutant), we identified five genomic regions that become specifically decompacted during maturation. These regions are characterized by low GC-content and are evolutionarily conserved across different adenovirus species, independent of protein-coding constraints. Adenoviral DNA packaging is guided by a distinct 6.1-bp dinucleotide periodicity pattern that helps position viral chromatin proteins. Core maturation serves a dual purpose: (i) it contributes to capsid uncoating by increasing internal pressure while (ii) simultaneously preparing the viral chromatin structure for rapid transcription upon nuclear entry. These findings reveal how sequence-encoded structural information guides adenoviral genome organization and suggest new approaches for optimizing therapeutical adenoviral vectors.**

**Keywords** Adenovirus; DMS-Seq; Bioinformatics; Virus Maturation
**Subject Categories** Chromatin, Transcription & Genomics; Microbiology, Virology & Host Pathogen Interaction

## Introduction

Adenoviruses (Ad) are non-enveloped, double-stranded DNA viruses causing self-limiting infections in immunocompetent hosts, while severe and life-threatening diseases are mostly restricted to immunosuppressed individuals (Hierholzer, 1992; Lion, 2014; Kajon, 2024). Their low pathogenic profile makes them versatile vector tools used in gene therapy (Gao et al, 2019; Zhang et al, 2018; Wold and Toth, 2013), oncolytic therapy (Cervera-Carrascon et al, 2019; Yamamoto and Curiel, 2010; Uusi-Kerttula et al, 2015), and vaccine development prominently seen during the Covid-19 pandemic (Mendonça et al, 2021). Understanding the structural changes driving the viral life cycle thus benefits both, combating adenoviral infections as well as developing their vector potential.

The C-type human adenoviruses (HAdV-C2/5) are prototypic members of the family. Their particles are formed as icosahedral capsids composed of the major capsid proteins, hexon, penton and fiber. These are stabilized through additional minor proteins and enclosing the ~36-kDa linear double-stranded viral DNA (Nemerow et al, 2012). The capsid plays an important role in the infection process by providing a temporary stable shell that protects the genetic material during the passage from the producer to the target cell, withstands the internal overpressure of the packaged genome and enables target cell recognition and genome delivery (Mangel and San Martín, 2014; de Pablo and San Martín, 2022). The genome itself is protected at each 5'-end by a covalently bound copy of the terminal protein (TP). It is condensed into chromatin made of the positively charged protamine-like core protein VII and the condensing protein X. The nucleoprotein core and the capsid are directly linked by a layer of protein V that interacts with core protein VII and X and the capsid interior through protein VI (Gallardo et al, 2021; Vayda et al, 1983; Pérez-Vargas et al, 2014; Rekosh et al, 1977). In addition, the viral particle contains few copies of protein IVa2 essential for genome packaging (Christensen et al, 2008; Ostapchuk et al, 2006) and several copies of the adenoviral protease required during virus maturation (Mangel and San Martín, 2014).

Adenoviruses enter cells through receptor-mediated endocytosis. This is followed by cytosolic escape from the endosome and retrograde transport to the nucleus. A key aspect of this process is the stepwise disassembly of the virus, driven by a series of structural changes that weaken the capsid, ultimately leading to core release at the nuclear pore for import into the nucleus (Greber et al, 1993; Pied and Wodrich, 2019; Greber and Suomalainen, 2022). To balance the need for particle assembly in the producer cell with the

[1]Department of Infectious Diseases/Virology, Medical Faculty, University of Heidelberg, BioQuant BQ0030, Im Neuenheimer Feld 267, 69120 Heidelberg, Germany. [2]Regensburg Center for Biochemistry, University of Regensburg, Universitätsstr. 31, 93053 Regensburg, Germany. [3]Faculty of Biosciences, University of Heidelberg, 69120 Heidelberg, Germany. [4]Leibniz Institute of Virology (LIV), Martinistraße 52, 20251 Hamburg, Germany. [5]Centre for Structural Systems Biology (CSSB), Notkestraße 85, 22607 Hamburg, Germany. [6]Microbiologie Fondamentale et Pathogénicité, MFP CNRS UMR 5234, University of Bordeaux, 146 rue Leo Saignat, CEDEX, 33076 Bordeaux, France. [7]Present address: Roche Pharma Research and Early Development, Therapeutic Modalities, Roche Innovation Center Basel, Hoffmann-La Roche Ltd., Grenzacherstrasse 124, 4070 Basel, Switzerland. ✉E-mail: harald.wodrich@u-bordeaux.fr; gernot.laengst@ur.de

need for efficient disassembly in the target cell, adenoviruses are initially produced as highly stable immature particles. Those are subsequently converted into mature particles with reduced stability, primed for the entry process. Virion maturation requires the adenovirus protease (AVP), which is incorporated into progenies to process six out of the thirteen structural proteins incorporated as pre-proteins; three minor capsid proteins located at the inner surface of the capsid (IIIa, VI, VIII) and three core proteins (VII, X, and TP), and the scaffold protein L2 52/55 K (Ostapchuk and Hearing, 2005; Mangel and San Martín, 2014).

Our current understanding of structural changes during adenovirus maturation and its impact on virus entry and genome delivery comes largely from studying the temperature-sensitive *ts1* mutant (Ad2/5-*ts1*). This mutant efficiently assembles a genome containing progeny but exhibits a maturation defect when grown at the non-permissive temperature due to a single point mutation (P137L) in the AVP, restricting its incorporation into the virion (Weber, 1976; Rancourt et al, 1995; Imelli et al, 2009; Martinez et al, 2015). Immature *ts1* particles retain hyperstability but are still capable of binding and entering target cells at rates similar to wild-type adenoviruses (Martinez et al, 2015; Imelli et al, 2009). Their capsids, however, fail to undergo the required partial uncoating process as seen with mature capsids. Receptor engagement and endocytic uptake normally trigger structural capsid alterations, resulting in penton removal and release of the membrane lytic protein VI from the capsid interior (Denning et al, 2019). One of the reasons for this disassembly defect is the lack of core protein processing. Cryo-EM and genetic studies showed competitive binding for pre-protein VI and pre-core protein VII for a binding site on the inner hexon surface. In the mature virion, the N-terminal peptide of protein VI occupies the site, showing how maturation drives infectivity by reorganizing both the capsid and the adenoviral core (Hernando-Pérez et al, 2020; Ostapchuk et al, 2017).

In the wild-type virion, processing results in reorganized cores, which creates internal pressure due to the repulsion forces of tightly packed DNA (Pérez-Berná et al, 2012; Ortega-Esteban et al, 2015). To address space and charge constraints, the major core proteins (V, VII, and X), which account for ~50% of the core's molecular weight, initially modulate the intrinsic repulsion forces through charge neutralization and structural organization, which allows genome incorporation into immature particles (Pérez-Berná et al, 2015; Martín-González et al, 2019a; Yu et al, 2022; Johnson et al, 2004). Upon maturation, processing of core proteins (VII, X, TP) alters the core structure, resulting in increased pressure, thereby controlling the stability of the processed capsid. Simultaneously maturation facilitates the release of membrane lytic capsid protein VI, which ruptures the endosomal membrane allowing the capsid to escape and access retrograde transport in the cytosol (Wiethoff et al, 2005; Moyer et al, 2011; Wodrich et al, 2010; Scherer et al, 2020; Bremner et al, 2009), culminating in core release at the nuclear pore complex (Pérez-Berná et al, 2012, 2015; Martín-González et al, 2019a; Gallardo et al, 2021). Similar effects destabilizing the virion are observed in the absence of protein VII, which also increases capsid pressure or when packaging shorter genomes (Hernando-Pérez et al, 2020; Smith et al, 2009).

To date, it is unknown how the processing of protein VII changes the genome structure to form an infection-competent virus, nor is the exact structure of the core known. Cryo-EM

tomography experiments suggest the presence of discrete nucleo-protein particles in the viral capsid, called adenosomes, being separated by stretches of loose DNA (Pérez-Berná et al, 2015). Releasing the nucleoprotein complex of the virion by pyridine treatment and electron microscopic visualization, reveals that the structure is similar to cellular nucleosomal arrays, also forming "beads on a string" like structures upon high salt treatment (Vayda et al, 1983). Using MNase digestion, we recently showed that during infection, ~200 pVII-DNA complexes protect the genome at defined positions. These positions are characterized by an underlying dinucleotide periodicity pattern suggesting a specific and sequence-driven genome organization during infection (Schwartz et al, 2023). Such beads are observed in wildtype but not pVII-deleted virions (Martín-González et al, 2019a), which agrees with previous studies suggesting about 200 pVII-DNA complexes with a diameter of 9.5 nm separated by linker DNA of varying length contributing to the core organization. This reconciles that 2–6 protein VII molecules may form a multimer, around which 30–150 base pairs of viral DNA are wrapped (Vayda et al, 1983; Vayda and Flint, 1987; Chatterjee et al, 1986; Mirza and Weber, 1982; Corden et al, 1976; Gyurcsik et al, 2006).

In this study, we employ DMS-seq (Umeyama and Ito, 2017) to compare the nucleoprotein structure of Ad5-wt and the Ad5-*ts1* mutant in the viral capsid (in virio). Dimethyl sulfate (DMS) penetrates the viral capsid and methylates purines at accessible, protein-free DNA sites. The quantitative evaluation of DNA methylation sites reveals a dramatic reorganization of the core during maturation. Sites of specific dinucleotide frequency nucleate the regular packaging by pVII protein prior to maturation. Upon maturation, this regular structure is specifically decompacted within five genomic regions with a low GC-content. These regions overlap with the early genes and one additional central region of the viral genome. Regions of chromatin decompaction, represent the sites of (cellular) nucleosome assembly upon nuclear entry, suggesting that a transcription-competent viral chromatin structure is already established during maturation and delivered to the target cell. Furthermore, we propose that chromatin decompaction domains are evolutionary conserved in Adenoviruses, generating a high internal pressure to allow capsid disassembly during infection. This finding provides evidence that evolutionary optimization of the viral DNA sequence influences structural organization, extending its role beyond mere genetic encoding.

## Results

### Ad5-wt and Ad5-*ts1* exhibit distinct DNA packaging

To study the viral nucleoprotein architecture in detail, we employed DMS-seq (Umeyama and Ito, 2017). Dimethyl sulfate (DMS) is a small molecule that methylates guanine (G) and adenine (A) in unprotected DNA. DNA methylations are converted to nicks and double strand breaks by a combination of spontaneous depurination, APE1, and Putrescine (Fig. EV1A,B; Material and methods). As DMS penetrates the viral capsid (Fig. EV1C), no additional manipulation of the virus is required. The DMS-treated viral DNA was isolated and used to prepare high-throughput sequencing libraries. Sequencing and mapping the DNA ends allowed us to map DMS-mediated cleavage sites (Figs. 1A and EV1). As DNA

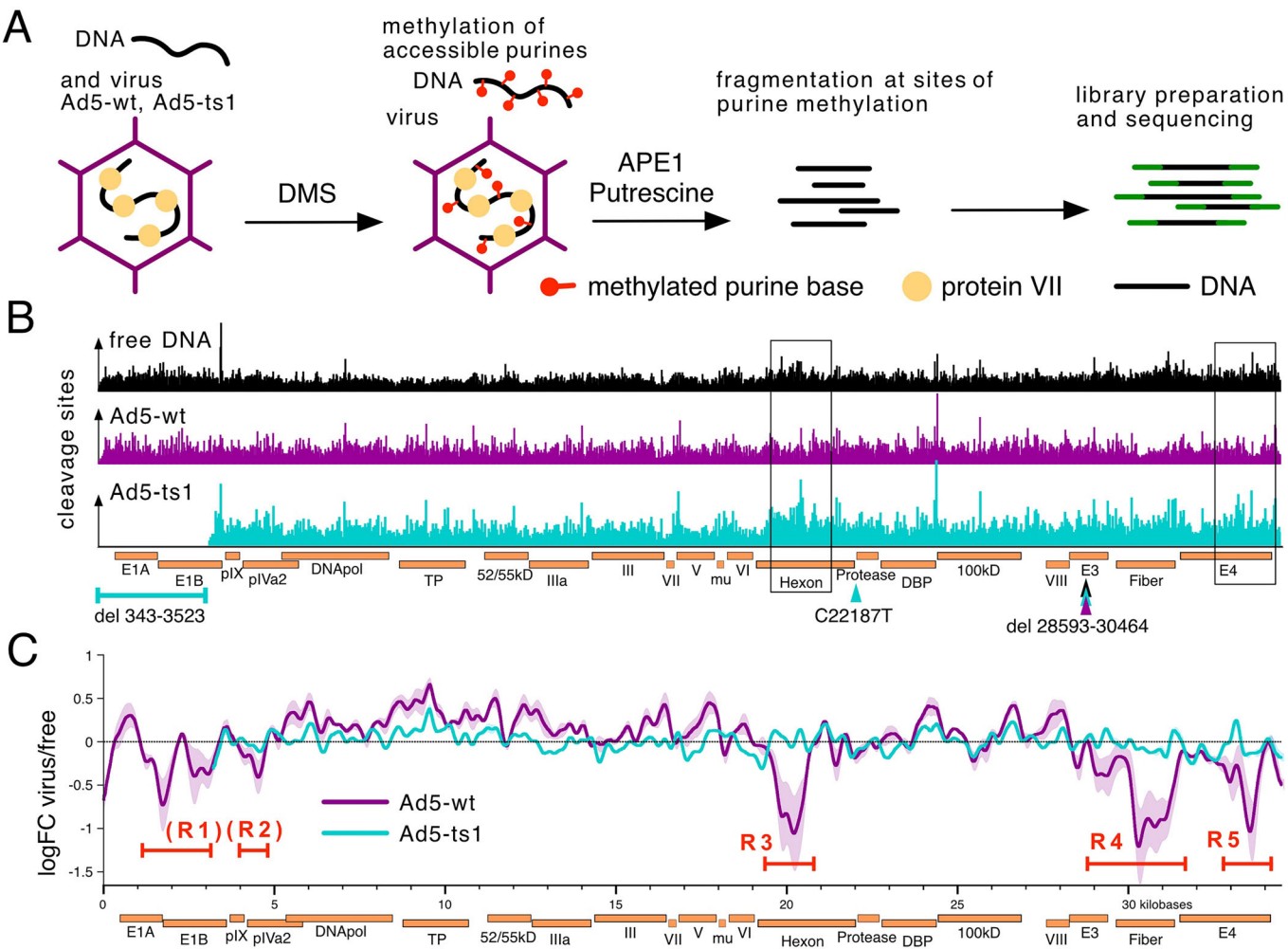

**Figure 1. DMS-seq and DMS-mediated cleavage frequencies alongside free DNA, Ad5-*ts1*, and Ad5-*wt*.**

(A) Overview of the DMS-seq procedure on adenoviral particles. DMS penetrates the viral capsid and methylates unprotected purines within. The DNA is purified and subsequently fragmented with APE1 and putrescine at the sites of DNA methylation. Fragmented DNA is subject to library preparation and high-throughput sequencing. (B) IGV representation (barplot) of DMS-induced cleavage events normalized to sequencing depth across the genomes of free DNA, Ad5-ts1 and Ad5-wt. Both Ad5-ts1 and Ad5-wt possess a partial E3-deletion. Ad5-ts1 has an additional deletion of 3180 bp in the E1A and E1B genes in addition to the C22187T mutation responsible for the mutant phenotype. Point mutations are indicated as arrows and deletions as brackets. Boxes point out two example regions in which the pattern on Ad5-ts1 substantially differs from Ad5-wt. (C) Log fold change (LogFC) of DNA methylation sites between virus samples and free DNA. The data is represented by a sliding window approach across the viral genomes. Five regions (R1–R5) with a notably decreased number of DMS-cuts in the Ad5-wt samples are indicated. The shaded area represents the range of the two biological replicates and the line shows the average value. Source data are available online for this figure.

methylation is guided by DNA-bound proteins, the method is capable of capturing the nucleoprotein structure of the virus in its native state.

Here we compare wild-type virus and a *ts1*-virus deleted for the E1 region, which for simplicity we call Ad5-*wt* and Ad5-*ts1* respectively (see Material and methods for details). High-throughput sequencing yielded between 27 and 33 million reads per sample, corresponding to an extraordinarily high 25,000- to 35,000-fold coverage of the viral genome (Table 1). As a control, protein-free Ad DNA was DMS-treated and sequenced to achieve a similar level of DNA fragmentation. All samples showed a similar size distribution of fragments with a peak at about 75 bp (75.5 bp free DNA, 72.5 bp Ad5-ts1, and 78.5 bp Ad5-wt; Fig. EV1D). To assess whether the DMS-mediated fragmentation worked as

intended, we quantified the nucleotide proportions at the 5'-end of the reads. As expected, the cleavage sites (position -1) consist of >90% purines (Fig. EV1F).

The 5'-ends of the fragments (cleavage sites) were mapped to their respective reference genomes, and counts were normalized to sequencing depth (Fig. EV1E). The resulting distribution of cleavage sites varies substantially between the samples. Ad5-wt possesses multiple regions with a lower number of cleavage sites compared to Ad-ts1 or free DNA (Fig. 1B, boxes). In addition, the corresponding correlation matrices show Ad5-wt to differ from Ad5-ts1 and free DNA (Fig. EV2A–C). In turn, the free DNA control is more similar to Ad5-ts1 than to Ad5-wt (Fig. EV2C,D). This similarity of Ad5-*ts1* and free DNA is also reflected in the log fold change of cleavage sites, being mostly close to 0 (Fig. 1C).

**Table 1. Overview of the samples, read depth and annotation rate.**

| Genotype | Name | Treatment | Number of reads | Alignment rate | Aligned reads | Coverage |
|----------|------|-----------|-----------------|----------------|---------------|----------|
| WT | free DNA R1 | DMS | 2.70E + 07 | 3.28% | 8.22E + 05 | 9.89E + 02 |
| WT | free DNA R2 | DMS | 3.20E + 07 | 2.59% | 7.76E + 05 | 9.35E + 02 |
| WT | Ad-wt R1 | DMS | 3.30E + 07 | 99.71% | 3.04E + 07 | 3.66E + 04 |
| WT | Ad-wt R2 | DMS | 2.40E + 07 | 99.42% | 2.23E + 07 | 2.68E + 04 |
| ts1 | Ad-ts1 R1 | DMS | 2.60E + 07 | 93.39% | 2.21E + 07 | 2.66E + 04 |
| ts1 | Ad-ts1 R2 | DMS | 2.90E + 07 | 93.50% | 2.55E + 07 | 3.06E + 04 |

However, the fold change of Ad5-wt DNA cleavage sites is greatly reduced in three genomic regions, termed R3 to R5 (Figs. 1C and EV2E). Two additional regions (R1 and R2) are marked where Ad5-wt patterns deviate from the free DNA pattern, albeit due to genomic deletions (deltaE1) of this region in our Ad5-ts1, we cannot be certain that these regions follow the same trend as R3 to R5. For this reason, we marked these sites with brackets as potential sites of deviation.

Fewer observed DNA cleavage sites could stem from either intense compaction and inaccessibility of underlying DNA or—conversely—from extremely low compaction and thus over-fragmentation of the highly methylated DNA. To investigate what might lead to the decrease in cleavage sites in R1–R5, we consulted V-plots showing the fragment size distribution along the viral genome (Henikoff et al, 2011). The V-plots show regions R1–R5 being covered by relatively short fragments, likely as a result of the DNA being particularly accessible to the DMS during library generation in those regions, leading to over-fragmentation and thereby loss of reads from small fragments (Fig. EV3A). Our recent MNase-seq analysis of viral DNA at early infection stages similarly suggested that several genomic regions exhibit lower compaction states (Schwartz et al, 2023). Indeed, the MNase-seq regions of low compaction do overlap with the regions R3 to R5 from DMS-seq (Fig. EV3B). This suggests the DMS data captures the same open chromatin structure of the viral genome—a feature that has already been established in the virion prior to cell entry.

## Arrays of phased adenosomes occupy the Ad genome

The close inspection of DMS-modified sites on viral DNA did not reveal a black and white picture of DNA protection, as expected by sequence-specific bound proteins (Fig. 2A). Instead, gradual changes of DMS modification could be observed over tens of base pairs, with apparent periodic peaks and dips in DMS accessibility (Fig. 2A, indicated by arrows). DMS modification is commonly used to analyze protein–DNA footprints and also to reveal nucleosome positioning (Umeyama and Ito, 2017; Shaw and Stewart, 2009). Such a pattern is reminiscent of arrays of well-positioned nucleosomes and may be the result of regularly spaced arrays of adenosomes (Kaplan et al, 2009; Segal et al, 2006). We recently showed that during early infection, the Ad5-wt genome is covered by regularly phased adenosomes by MNase-seq experiments (Schwartz et al, 2023). To reveal a potential periodic occurrence of DMS cleavage sites on the viral DNA, we leveraged spectral density estimation (SDE), a method that was previously used to reveal the phasing of nucleosomes (Baldi et al, 2018). The resulting SDE shows a relatively broad peak between periods 70 and 100 bp for all samples, including the free DNA (Fig. 2B). This indicates that DMS cleavage sites in our samples have the tendency of appearing periodically about every 85 bp. To exclude the impact of DNA fragment length on periodicity, we consider only one random end of a DNA fragment to calculate the DMS footprint and periodogram, serving as a control. This exhibited negligible differences, suggesting that specific viral DNA packaging infers periodicity, and DNA fragmentation length has a minimal impact. Surprisingly, the periodicity of DMS cleavage sites on free DNA is very similar to those of the virions, albeit with strongly reduced intensity (Fig. 2B). DMS-seq thus seems to capture a sequence-dependent property of the DNA itself. This property is strongly enhanced in the presence of viral DNA-binding proteins, suggesting that the DNA-encoded signal serves as a template for organizing the nucleoprotein structure.

As additional control, we calculated the SDE of the publicly available DMS-seq data of a *Saccharomyces cerevisiae* study (Umeyama and Ito, 2017). Here as well, we observe a broad peak in free DNA with a period of about 150 bp, matching the size of a nucleosome (Luger et al, 1997). However, the yeast SDE for chromatin is remarkably weaker than the viral periodicity, suggesting a higher regularity of viral DNA packaging.

To reveal genomic regions that contribute most to the genome-wide SDE (Fig. 2B), we generated heatmaps, plotting the spectral density of periods between 50 and 130 bp in sliding windows across the entire viral genome (Fig. 2D). Visual inspection of free DNA and viral heatmaps shows clear similarities. However, the SDE signal is strongly enhanced in the viral nucleoprotein structure, with additional sites of strong periodicity. The stronger viral signals are likely due to a more defined, regular spacing of the adenosomes (higher power spectral density values), without broadening these regions (Fig. 2D,E). Ad5-wt exhibits one strong SDE signal in the center of the genome (genome position 24.000), whereas Ad5-ts1 has three additional strong periodicity centers, marked by arrows (Fig. 2D,E). Interestingly, the additional Ad5-ts1 sites overlap perfectly with the open regions of Ad5-wt (regions R3 to R5, and to a lesser extent R2 in Fig. 1). Besides these peaks of high regularity, the overall similarity between the spectral density heatmaps of free DNA and viral samples is shown by calculating the Pearson correlation (free DNA to viral signals: 0.72 Ad5-ts1, 0.54 Ad5-wt; Ad5-ts1 to Ad5-wt signal: 0.71; Fig. 2C). The similarity of free DNA and virus periodicity patterns suggests that underlying DNA-encoded patterns might serve as nucleation points for the organization of DNA packaging, with the bound proteins strengthening the periodic pattern (Fig. 2E).

Remarkably, the genome-wide DMS accessibility pattern of free DNA and Ad5-ts1 is very similar to each other, except for the

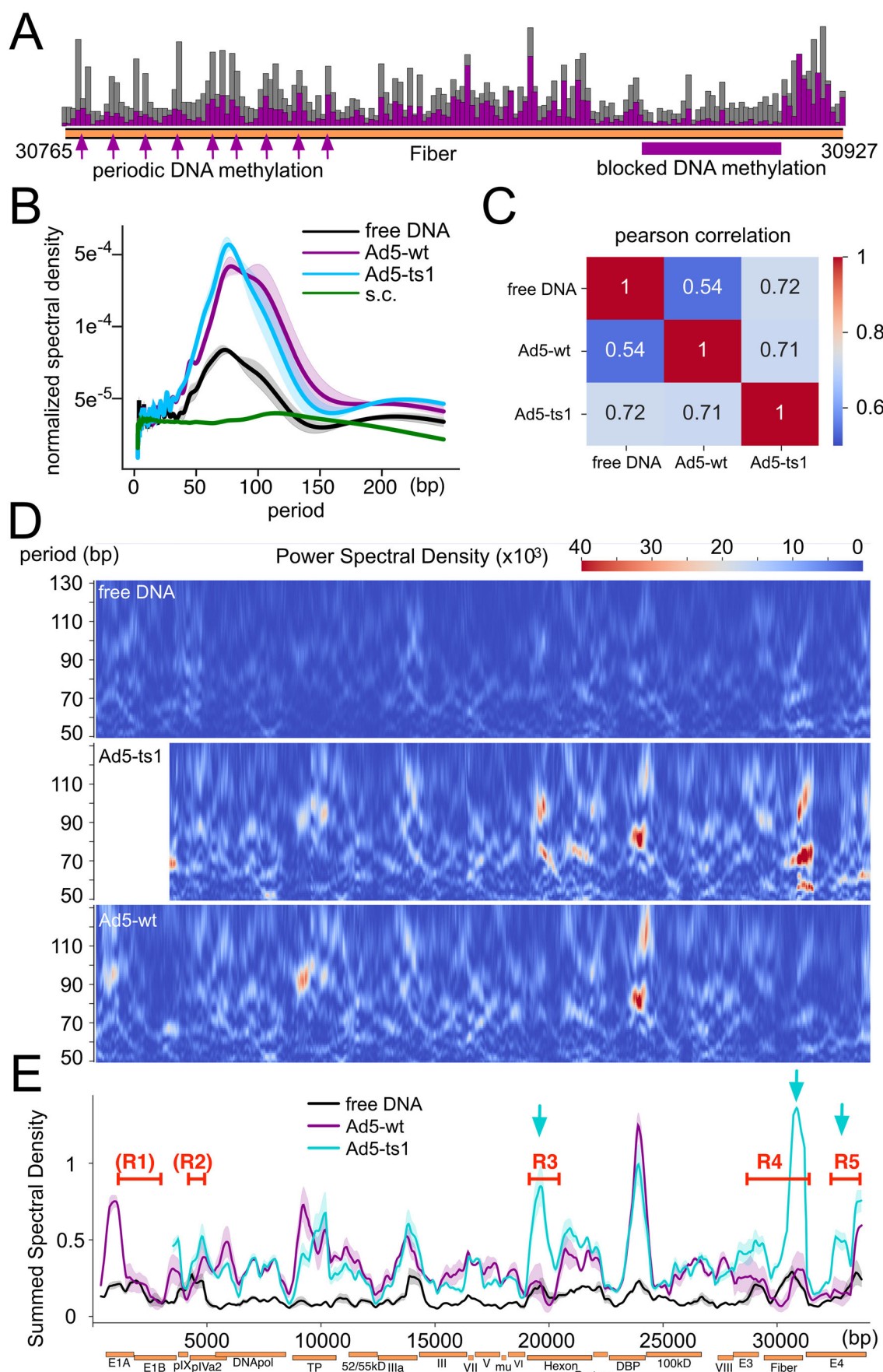

**Figure 2. Periodic DMS methylation patterns in free DNA and viral samples.**

(A) Example of a zoomed-in section of the genome, with every bar indicating one base pair. The DNA methylation pattern indicates a periodic pattern of increased cut sites in certain regions of the genome, present in the viral samples and absent in the free DNA (indicated by arrows). Other genomic regions exhibit larger domains of evenly reduced DNA methylation (indicated by a horizontal bar). (B) Welch Periodogram of the DMS-induced DNA cleavage site distribution in the replicates of each sample. This plot shows the power spectral density (PSD, y-axis) of the periodicity (x-axis) of a signal; in other words, it shows the periods that have the highest impact in explaining the data. The viral samples show a high periodicity for periods between 70 and 100 bp. A lower peak can be identified in the free DNA samples. The yeast samples (*Saccharomyces cerevisiae* BY4741 (ID 408084) - BioProject - NCBI) show a peak at around 130 bp periods. (C) Pearson correlation matrix of the heatmaps. (D) Heatmaps of pooled replicates showing the Power Spectral Density (color gradient) of specific periods (y-axis) in bins across the entire genome (x-axis). For this plot, the number of DMS-induced DNA cleavage sites were divided into 750 bp bins with a step size of 35 bp. Only the relevant periods of 50–130 bp are shown in the heatmaps. Heatmaps of the replicates were averaged. (E) The spectral density of all periods in a genomic bin was summed up to collapse the heatmaps into a line plot to allow for better visibility of highly periodic regions. Regions R1–R5 with a decreased number of DMS-dependent DNA methylation sites in Ad5-wt are indicated. The latter 3 (R3–R5) coincide with areas of increased periodicity in Ad5-*ts1*, but not in Ad5-*wt* (indicated by arrows). The shaded area represents the range of the two biological replicates and the line shows the average value. Source data are available online for this figure.

regions of DNA decompaction in Ad5-wt (Fig. 1C). However, the spectral density map of Ad5-*ts1* is more similar to Ad5-wt than to free DNA (Fig. 2B–D), suggesting that additional DNA-encoded signals induce the opening of the regions R3 to R5 (Fig. 1C). In addition, the spectral density analysis showed that (i) the nucleoprotein complexes of Ad are phased at an interval between 70 and 100 bp, (ii) suggesting that like for eukaryotic DNA packaging, the existing periodicity in the viral DNA sequence guides adenosome positioning, and (iii) that maturation dependent changes in DMS cleavage periodicity are limited to the genomic regions R1–R5 (Kaplan et al, 2009; Segal et al, 2006).

## Conserved dinucleotide patterns determine adenosome periodicity

In eukaryotic genomes, the 10–11 bp periodical distribution of dinucleotide motifs was described as a DNA-encoded nucleosome positioning signal (Kaplan et al, 2009). Sequence motifs determine the specific structure of DNA, altering the canonical helical path of the DNA (Rohs et al, 2009; Abe et al, 2015). For example, it is known that GC dinucleotides form wider major grooves, whereas AT or AA dinucleotides come with a narrower major groove. The human genome exhibits a 10 bp periodicity of GC dinucleotides and an additional 10 bp periodicity of AT or AA shifted by 5 bp relative to the GC frame, resulting in curved DNA sequences (Kogan et al, 2006). The curved DNA has reduced energy costs for nucleosome formation and positions nucleosomes in vivo (Bettecken et al, 2011).

The occurrence of DNA methylation periodicity on free DNA (Fig. 2) led us to analyze the dinucleotide pattern of Ad5. We detected a striking periodicity of YY (T and/or C) and RR (A and/or G) dinucleotides with a repeat length of 6.1 bp that was not detectable for other dinucleotide motifs (Figs. 3A and EV4B). The periodic intensity for the adenovirus sequences is more prominent than the AA or TT dinucleotide signals guiding nucleosome positioning in yeast (Fig. EV4A). Shuffling the Adenoviral genome sequence results in a loss of this repeat pattern. (Fig. EV4C). Furthermore, we analyzed additional Adenoviral and other dsDNA virus genomes to test whether this is a general principle. No 6.1 bp periodicity can be detected in smallpox virus, but the human Adenovirus types 4 and 2 show a similar, pronounced periodicity of YY and RR dinucleotides (Fig. 3A). The yeast genome exhibits the 10 bp periodicity also seen in other eukaryotes. The more distant bat Ad2 genome exhibits a YY periodicity, but the RR repeat signal

is not present. The analysis shows that periodicities are a common phenomenon in Adenoviruses, suggesting a functional role.

Next, we asked whether the genomic dinucleotide periodicity of 6.1 bp overlaps with the periodic packaging patterns of Adenovirus (Fig. 3A). To quantify the periodicity of the dinucleotides YY and RR across the genome, we visualized the spectral density in a heatmap from periods 4 to 20, similar to how the DMS methylation pattern was processed (see Methods and Fig. 2C). Dinucleotide periodicities are not randomly distributed along the viral DNA, but cluster in certain genomic regions. Interestingly, these regions overlap with the pronounced periodic DNA methylation patterns of viral DNA (Fig. 3B,C). In order to visualize this dependency, the periodicity of YY and RR dinucleotides was summed up into a line graph and drawn together with the periodic nucleoprotein patterns of Ad5-*ts1* and Ad5-wt (Fig. 3B). Quantification of the overlap of dinucleotide periodicity peaks with those of periodically positioned adenosomes is given in Fig. 3C. The plot confirms a highly significant co-localization of dinucleotide frequency peaks with the periodic DNA methylation pattern, suggesting that the dinucleotide pattern plays a role in guiding adenoviral DNA packaging.

## The local GC-content correlates with domains of genome decompaction upon maturation

YY and RR dinucleotide periodicities are potentially guiding the adenovirus nucleoprotein structure, defining domains of regular adenosome array formation. However, the DNA-encoded signal(s) driving DNA decompaction of the regions R1 to R5 upon proteolytic maturation remain unclear. Addressing additional sequence properties we identified a high correlation between the local GC-content of the genome and the maturation dependent genome opening (Fig. 4A). The pattern of the GC-content clearly follows the quantitative DMS cleavage differences of Ad5-wt relative to free DNA and Ad5-*ts1*, as displayed by the logFC plot. Low GC levels perfectly mirror the DMS-dependent nucleoprotein structure of the wild-type virus. The regions R1 to R5, displaying the DMS accessible adenovirus regions, do perfectly match with the genomic domains of low GC-content.

Ad5-wt decompaction regions correlate with the GC-content, whereas an anti-correlation is observed for free DNA and Ad5-*ts1* (Fig. 4B, left panel). The relative DMS cleavage differences between the viruses and free DNA (logFC values from 1 C), plotted with the GC-content, also show a correlation of the Ad5-wt nucleoprotein

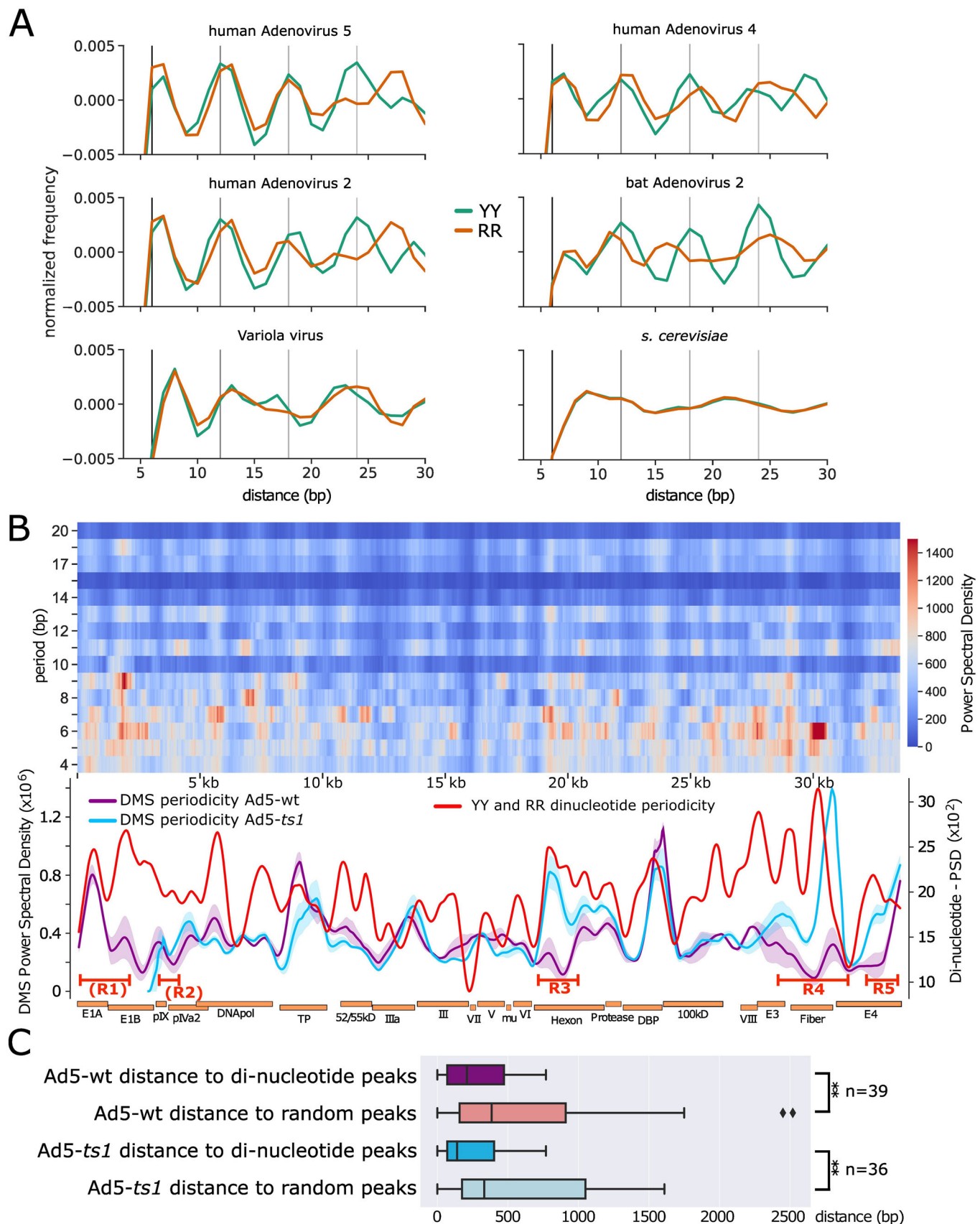

**Figure 3. Dinucleotide repeat patterns along the adenoviral genome.**

(A) Distance histograms for YY (T/C) and RR (A/G) dinucleotides for the indicated genomes. Adenoviral genomes display a repeat pattern with a period of about 6 bp, which is absent in smallpox (Variola major) and in yeast (S. cerevisiae). Vertical lines at a distance of 6 bp help to visualize the repeat pattern. (B) Power spectral density of the dinucleotide patterns YY and RR in human adenovirus 5 as a heatmap, similar to Fig. 2C. Only the periods between 4 and 20 are shown. To indicate genomic areas with a strong repeat pattern, the heatmap from periods 5–7 is summed up into a line plot and smoothed with lowess. The periodicity of Ad5-wt and Ad5-ts1 was smoothed in a similar way and plotted in the same graph. The shaded area represents the range of the two biological replicates and the line shows the average value. (C) Boxplot representing the distance of every DMS periodicity peak (Fig. 2) to the closest peak in the dinucleotide periodicity (red line). As a control, a set of randomly positioned peaks was generated (the same number as there are dinucleotide peaks), and the distances to these peaks were also measured and compared. The boxes extend from the first to the third quartile, with the median shown. The whiskers reach either 1.5 times the interquartile range or the maximum data value, whichever comes first. Data points 1.5 times beyond the interquartile range are shown as outliers. For statistical analysis, a two-sample $t$-test was performed for the Ad5-wt ($p = 0.0023$) and Ad5-ts1 comparison ($p = 0.0022$). Samples with a $p \leq 0.01$ are indicated by "**". Source data are available online for this figure.

structure with the GC-content ($r = 0.76$), but not for Ad5-ts1 (0.18) (Fig. 4B, right panel).

In summary, the data suggest that the domains of low GC-content do not disturb the initial association of DNA-binding proteins with the Adenoviral genome, as the DMS methylation pattern of Ad5-ts1 mirrors the pattern of free DNA. However, upon protease cleavage, the processed proteins covering low GC-content regions undergo a structural change, or are displaced, adopting a new chromatin structure resulting in increased DNA accessibility.

Next, we explored whether low GC-content and viral decompaction domains are functionally dependent. For this, we addressed the evolutionary conservation of the low GC-content domains in the human Adenoviruses C5, B7, F41, D26 and the bat Adenovirus genome (Fig. 5A; other examples in Fig. EV5). Plotting the data on the relative genomic scale revealed that all genomes exhibited domains of low GC-content and a similar distribution of low and high GC-content regions. The genome ends are characterized by a low GC-content in addition to 1 to 2 pronounced regions of GC reduction in the central region of the genomes. A pattern that can also be observed in the bat virus. Both B7 and F41 are characterized by an additional GC dip, present in the penton region. By association, we predict that this region represents an additional nucleoprotein decompaction center, when compared to C5. Furthermore, D6, B7 and F41 exhibit a broader R4 region. The conservation of the low GC-content domains suggests that even among diverse adenoviruses, the structural reorganization of the genome upon protease cleavage may be conserved, and even more pronounced in other adenovirus species.

### Low GC-content domains are not determined by conserved protein sequences

One possibility that may lead to conserved low GC-domains amongst the viruses is that underlying protein or regulatory sequences define a functional need for this kind of base composition. This effect can be ruled out for the low GC-content regions in B7 and F41 that do not have a genomic counterpart in C5. Regions R3 and R4 appear in the coding regions of the Hexon and Fiber proteins of Ad5-wt, respectively. To rule out the protein-coding sequence as the reason for a similar base composition, these regions were analyzed in more detail. R3 is located within the first half of the hexon protein-coding sequence. We overlaid the percentage of sequence variability from a multiple sequence alignment between the analyzed viral hexon proteins with the GC-content (Fig. 5B). The low GC-content region belongs to the protein domains of highest sequence divergence, clearly showing

that GC levels do not follow a specific translation requirement for protein function. Furthermore, for AdB7 and AdF41, the low GC region is expanded to the second half of the protein, suggesting the GC-content is not the major driver of hexon function. The same results were obtained with region R4, which is larger and fully encompasses the Fiber gene product. Fiber is highly divergent between virus species, and low GC-content regions are associated with higher protein diversity (Robinson et al, 2013). Furthermore, the location and broadness of the low GC-content domains are shifted and split into subdomains, when comparing AdC5 and AdD26 sequences, suggesting again that protein sequence does not dominate the local GC levels. These observations argue against a protein-dependent selection pressure on DNA sequence, suggesting that the low GC-domain could play a specific function in organizing the viral nucleoprotein structure.

## Discussion

Adenovirus infectivity critically depends on the maturation process mediated by the adenoviral protease (AVP), which facilitates structural rearrangements essential for virus entry and genome delivery. While structural details of the maturation process have been obtained for the capsid by comparing immature TS1-mutant capsids with their wild-type counterpart (Yu et al, 2022), structural insights into changes in the viral core upon maturation remain unclear despite proteolytic processing of all three core proteins. This is also in part due to limitations in structural approaches that can reveal structural details in the absence of symmetry. To overcome this limit, we applied DMS-seq to investigate core changes in high resolution.

The comparison of free DNA, the Ad5-ts1 mutant and Ad5-wt virus DMS-seq data shows a substantial reorganization of the adenoviral core during its maturation. DMS-seq analysis revealed that a homogenous DNA packaging in Ad5-ts1, guided by dinucleotide patterns, dramatically changes upon AVP protease cleavage. Specifically, five domains with low GC-content acquire increased DMS sensitivity, suggesting a specific decondensation of these genomic regions. We show that the evolutionary conserved low GC-content regions in different adenoviral genomes are not a consequence of the underlying protein-coding sequence, but independent domains. These domains are most probably shaping the mature viral core structure, required for efficient host infection.

DMS-seq presents a high-resolution method to study DNA structure and protein–DNA interactions. DMS-dependent DNA methylation is guided by the DNA sequence/structure and is small

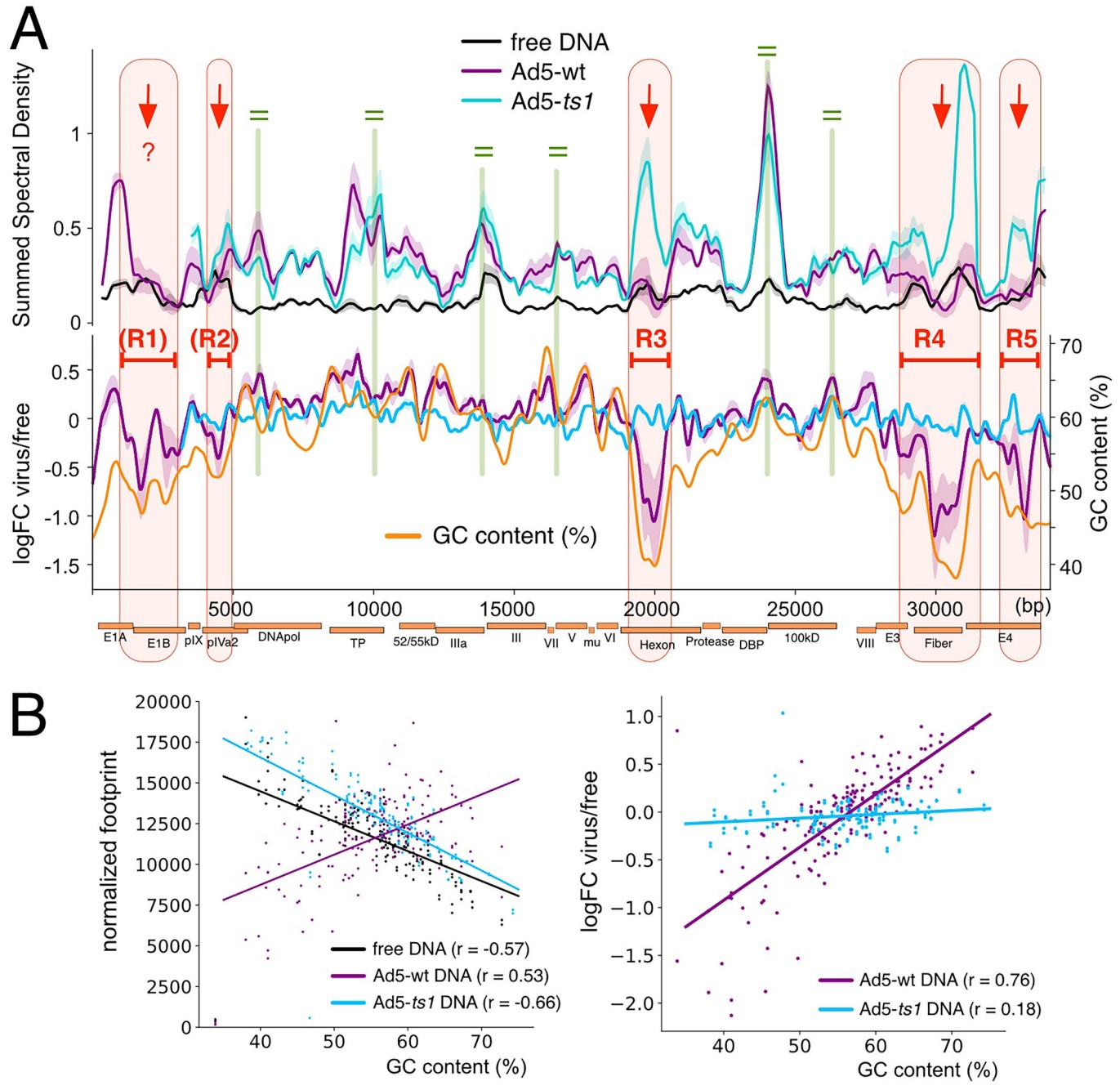

**Figure 4. Low GC-content mirrors the regions of reduced DMS-dependent DNA methylation on Ad5-wt.**

(A) The summed spectral density of the DNA methylation pattern in free DNA, Ad5-ts1, and Ad5-wt (upper panel) is shown together with the log fold change of free DNA methylation to viral DNA DMS-cuts (lower panel). The lower plot includes the GC-content of the genomic region (yellow curve) plotted in addition to the DNA methylation data, as indicated. The shaded area represents the range of the two biological replicates and the line shows the average value. The regions R1 to R5 with decreased DMS cleavage sites in Ad5-wt are indicated. Red arrows depict regions with decreased DNA methylation of the Ad5-wt compared to Ad5-ts1. Green equal marks depict regions with no change of DNA methylation between both viruses. (B) Dotplot of genomic bins comparing the sequencing depth normalized DMS-cuts with GC-content (left) and the logFC of free DNA methylation relative to the viral (DNA methylation with GC-content (right). Regression lines are drawn, and Pearson's r was calculated. Source data are available online for this figure.

enough to penetrate the viral capsid as well as DNA-bound proteins, only being blocked by direct amino acid - DNA interactions (Umeyama and Ito, 2017). DMS-seq has been shown to reveal site-specific binding of transcription factors to DNA and also resolve positioned nucleosomes (Umeyama and Ito, 2017). As DMS-seq is an ensemble approach, overlaying methylation signals of many genomes and variable protein positions on DNA, are not resolved by a discrete footprint, but by a variation of DNA

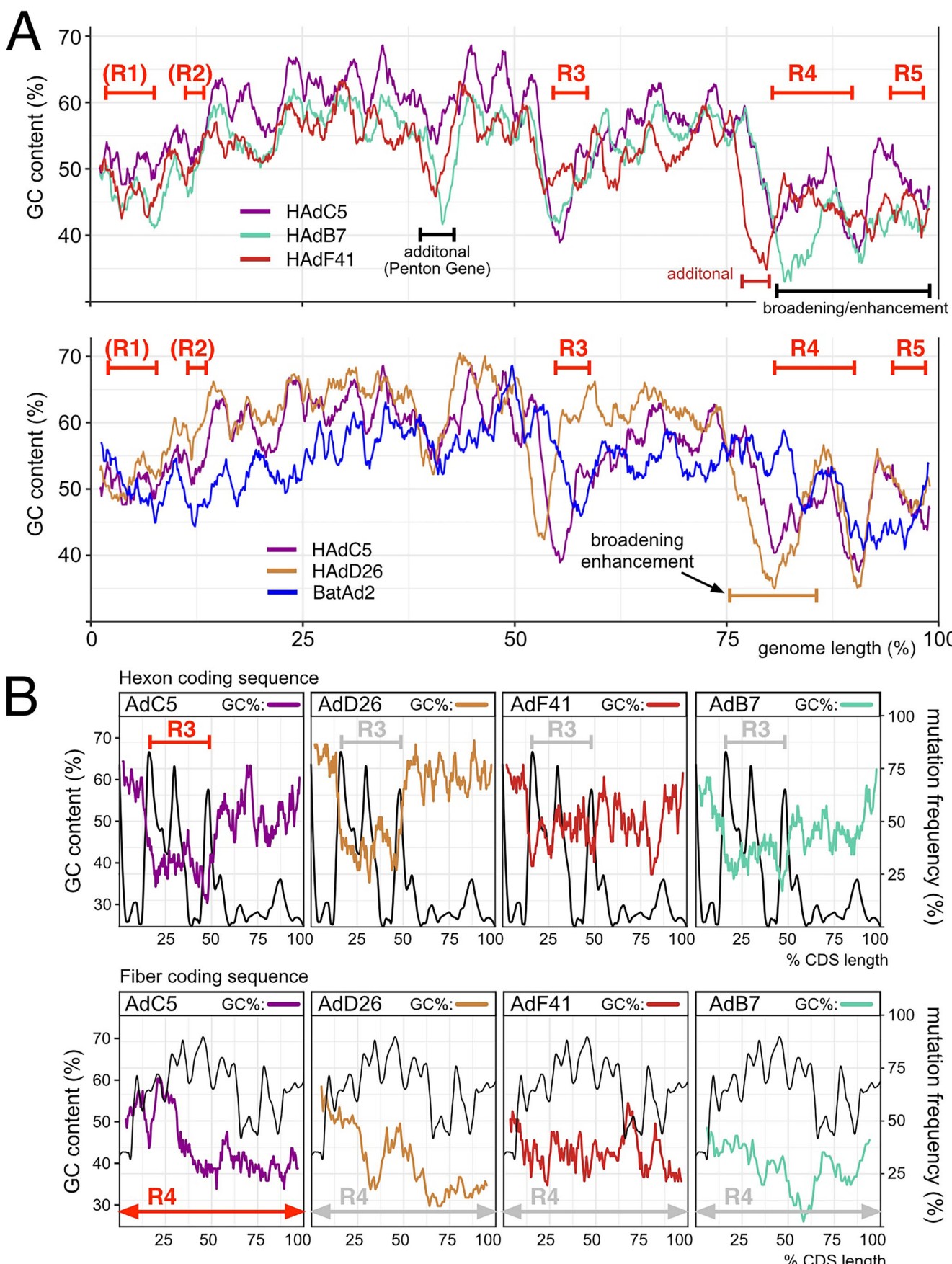

**Figure 5. Evolutionary conservation of genomic features.**

(A) Binned GC-content across the genomes of the indicated adenoviruses. Because of the different genome sizes, the position is given in percent of genome length. The GC-content of the indicated viruses are plotted together with the Ad5-wt studied here that corresponds to full-length HAdC5. For clarity, each plot shows only two additional Ad genomes. All of the depicted adenoviruses have GC-content dips with varying widths in similar positions of the genome. Additional GC-low regions, or broadened GC-low regions of the analyzed viruses relative to HAdC5, are indicated. (B) Detailed analysis of the GC-low regions within genes of the different Ad viruses and plotting the site-specific protein mutation rates between adenoviruses. The corresponding regions (R3 and R4) that encompass these GC-low regions are shown. Source data are available online for this figure.

methylation levels at these sites compared to the control. DNA methylation of protein-bound DNA can also increase compared to free DNA control, when protein–DNA interactions change the geometry of the major groove and the accessibility of the A and G bases (McGhee and Felsenfeld, 1979).

Using DMS-seq, we identified five regions (R1–R5) with increased global DMS accessibility in fully mature Ad5-wt in comparison to immature Ad5-*ts1* virions or free viral DNA (Fig. 1). The decompacted regions R1 to R5 are located close to the ends of the viral genome (R1, R2, R4, R5) and more central, overlapping with the hexon gene (R3). The terminal regions being accessible overlap with early genes that are expressed shortly after infection from the incoming genome. This observation suggests that those regions are primed for host protein binding and thus prepared for rapid transcription. In fact, previous MNase-seq data on the accessibility of the Ad5-wt genome during infection support this conclusion (Schwartz et al, 2023). The MNase-accessible regions are bound by cellular nucleosomes after nuclear entry, replacing viral adenosomes composed of protein VII. We propose that the overall structural organization of a transcription-competent viral genome is established during viral maturation.

The opening of the late genes hexon and fiber in regions R3 and R4, which are only transcribed following genome replication, points to a secondary function of Ad maturation of the core in capsid stability. Atomic force microscopy experiments have shown that the physical properties of the virion are influenced by core maturation, because (i) a mature viral core is necessary to destabilize and remove penton from the capsid which facilitates protein VI release (Denning et al, 2019) and (ii) Ad5-wt matured cores possesses increased internal pressure compared to the immature virion (Ortega-Esteban et al, 2015; Martín-González et al, 2019a). The latter studies show that processing of the genome-associated proteins X and VII by AVP, reduces DNA compaction in the core, resulting in increased internal pressure due to higher levels of electrostatic repulsion and DNA bending. We describe here the genomic regions and the encoded signals therein driving nucleoprotein decompaction, which may drive an increase in core pressure required to assist the subsequent capsid disassembly upon entry, for endosomal escape and genome release (Martín-González et al, 2019a).

Next to the global DMS-seq analysis, local DNA methylation differs at base pair resolution (Fig. 2). We identified several hotspots of well-positioned adenosomes that are guided by specific dinucleotide patterns and also partially overlap with the regions of DNA decompaction. The spectral density analysis shows a similar pattern on protein-free DNA and packaged viral DNA, albeit being less pronounced in the latter (Fig. 2B,C). A similar effect was observed in the DMS-seq study of Umeyama and Ito (Umeyama and Ito, 2017). DMS is modifying free DNA similarly to nucleosomal DNA, albeit to a lesser extent. The authors suggested that the natural shape of nucleosome-binding DNA gives rise to

this observation. This phenomenon is likely intertwined with the ~10 bp repeat pattern of dinucleotides that is detectable in many eukaryotic genomes and which is suggested to guide the positioning of nucleosomes in vivo (Segal et al, 2006). Likewise, adenoviral DNA exhibits a strong dinucleotide repeat pattern for YY (T/C) and RR (A/G) repeating every 6.1 bp, which is notably different from the ~10 bp observed in most eukaryotes. In our previous work, we identified a slightly different 5.4 bp SS/WW pattern centered on pVII peaks of invading Adenoviruses (Schwartz et al, 2023). These patterns are likely not functionally distinct, and their difference can be explained by the employed methods. Here, we analyze patterns along the entire viral genome, whereas in the previous study, we monitored only the local sequence elements bound by a single pVII (adenosome). The dinucleotide patterns are located in regions with high DMS signal periodicity (Fig. 3C), suggesting that the dinucleotide pattern might act as nucleation points for adenosome formation and positioning relevant for the assembly of viral chromatin prior to maturation, when positioning of viral proteins on the genome serves a higher net-neutralization (Martín-González et al, 2019b).

Besides the involvement of dinucleotide patterns in adenosome positioning, we observed that decompaction regions R1–R5 are characterized by a low GC-content. The conservation of the low GC-content regions in other Adenoviruses shows that domains of low GC-content are a general feature, suggesting a functional role. We show that the initial association of DNA-binding proteins with the Ad5-*ts1* genome is not disturbed by the low GC-content. However, the AVP-dependent proteolytic cleavage of protein X and VII results in the reorganization of the nucleoprotein structure, thereby increasing DNA accessibility.

Based on our analysis, we offer an extended model for the viral maturation process. We propose that after encapsidation of adenovirus DNA, the dinucleotide pattern together with GC-content guides the formation of adenosomes, resulting in charge-neutralized and compacted genomes. Upon cleavage of protein VII and protein X, the core is not loosened uniformly. Instead, subregions of the viral genome, in particular R3 and R4 but also R1, R2, and R5, adopt an altered structure and divide the genome into (DMS) accessible and non-accessible subregions, ultimately leading to higher electrostatic repulsion across the packaged genome. Because we find that this structural reorganization of the viral genome is maintained throughout nuclear genome delivery, we suggest that core maturation serves a dual purpose. Firstly—complementary to the maturation of the capsid—the maturation of the core increases internal pressure, which helps capsid disassembly upon entry as previously suggested (Martín-González et al, 2019b). Secondly, as a major finding of our work, we show that the whole genome enters the nucleus in a "transcription" conducive state in which early transcribed genes are open to become populated with

host proteins to drive viral gene expression (Schwartz et al, 2019). From the viral point of view, this would greatly shorten the time between genome delivery and establishing the gene expression program during which viral genomes are vulnerable to cellular countermeasures.

On top of the coding regions necessary for the viral life cycle, our results show that the Ad genomic sequence contains additional functionalities implicated in its maturation process—the YY/RR repeat pattern for adenosome formation and GC-low regions for decondensation. Understanding how these sequence features act together might be pivotal for the design of efficient adenoviral vectors for vaccine or gene therapy in the future.

# Methods

### Reagents and tools table

| Reagent/ resource | Reference or source | Identifier or catalog number |
|---|---|---|
| **Experimental models** | | |
| Ad5-ts1 | Harald Wodrich, Martinez et al, 2015 | |
| **Recombinant DNA** | | |
| Ad5-wt | Harald Wodrich, Martinez et al, 2015 | |
| Ad5-ts1 | Harald Wodrich, Martinez et al, 2015 | |
| **Chemicals, enzymes and other reagents** | | |
| DMS | Sigma-Aldrich | Sigma-Aldrich-D186309 |
| Putrescine | Sigma-Aldrich | Sigma-Aldrich-P5780 |
| APE1 | NEB | M0282S |
| **Software** | | |
| FastQC | https://www.bioinformatics.babraham.ac.uk/projects/fastqc/ | v0.11.8 |
| trimmomatic | Bolger et al, 2014 | V0.39 |
| bowtie2 | Langmead and Salzberg, 2012 | V2.4.1 |
| samtools | Li et al, 2009 | V1.10 |
| picard | Picard toolkit, 2019 | v2.21.8 |
| deeptools | Ramírez et al, 2016 | V3.5.0 |
| homer | Heinz et al, 2010 | V4.11.1 |
| bedtools | Quinlan and Hall, 2010 | |
| SciPy | Virtanen et al, 2020 | V1.3.1 |
| seaborn | Waskom, 2021 | |
| **Other** | | |
| NextSeq 2000 | Illumina | |

### Adenovirus sequences

For this study, a partially E3-deleted (del 28593-30464) but replication-competent human adenovirus serotype 5 was used and is referred to in this study as Ad5-wt. The free DNA control consisted of the same complete, partially E3-deleted (del 28593-30464) wild-type adenovirus sequence incorporated into a bacterial artificial chromosome (BAC, described in Schwartz et al, 2023). The sequence and annotation were downloaded and adjusted for the E3-deletion (accession: AY339865.1). The Ad5-ts1 mutant used in this study contains the same deletion at E3 as Ad-wt, additionally a replacement of the E1 region with a CMV mCherry expression cassette and the P137L mutation in the L3/p23 protease responsible for its phenotype (described in (Martinez et al, 2015)). It was grown at non-permissive temperatures. For annotation, the Ad5-ts1 genome uses the same sequence as Ad5-wt adjusted for mutations and deletions.

All viruses and vectors were amplified in E1 trans-complementing HEK293 cells and purified by double banding on CsCl gradients, extensively dialyzed against phosphate-buffered saline (PBS)-10% glycerol, and stored in aliquots at $-80\,°C$.

## DMS fragmentation

Dimethyl sulfate (DMS) is a small molecule that methylates guanine (G) and adenine (A) in free DNA. Every purine base on either strand of DNA serves as a potential target of DMS. Methylated purines have a highly increased rate of spontaneous depurination, leaving apurinic sites (AP-sites). The resulting apurinic sites (AP-sites) can be excised by the addition of putrescine and the endonuclease APE1. Putrescine is a diamine that acts as a catalyst for the β-elimination of the remaining sugar in an AP-site. APE1 hydrolyzes the phosphor-ester bond of an AP-site at its 5' end (McHugh and Knowland, 1995; Nakamura and Swenberg, 1999). A DNA double-strand break will only occur when the excision event also occurs on the reverse strand (Fig. EV1A).

The DMS reactions were established on free DNA, to identify conditions that partially fragment DNA to sizes between 200 and 500 bp, as described in (Umeyama and Ito, 2017). DNA was incubated with 5% DMS for 2 to 8 min, subsequently de-purinated, cleaved and purified for its evaluation by agarose gel electrophoresis (Fig. EV1B). The same conditions were used to methylate a bacterial artificial chromosome containing the Ad5 DNA (1 ug of DNA) as well as $2.7 \times 10^{10}$ Ad5-wt and $2.7 \times 10^{10}$ Ad5-ts1 virions (corresponding to 1ug of DNA). Virions were resuspended in 300 μL water and then incubated with 2% DMS. After 8 min the 100 μL reaction mix was added to 100 μL pre-prepared Stop solution containing 1 M beta-mercaptoethanol, 0.6 M sodium acetate and 0.12% glycogen. The methylated virions were incubated in 1% SDS and 1 μg Proteinase K (5 ng/μL) at 50 °C for 1 h. All methylated DNA samples were then precipitated with isopropanol, washed twice in 70% ethanol and the pellet dried at room temperature. The pellet was resuspended in water and the cleavage mix, consisting of 0.2 U/μL APE1, 50 mM Putrescine, and 1xNEBuffer 4. The cleavage reaction was incubated overnight at 37 °C. Because Putrescine can interfere with the gel-running properties of DNA due to its positive charge, the DNA from the cleavage reaction was ethanol precipitated, the pellet was washed once in 70% ethanol, and then dried at room temperature. The Pellet was resuspended in 20 μL of water. For initial analysis of the size distribution of fragmented DNA, 10 μL DNA was run on a 1.3% agarose gel at 125 V for 40 min and then stained using Ethidium Bromide (Fig. EV1C). One μL of the remaining 10 μL of fragmented DNA was also analyzed on an Agilent Tapestation, to

make sure the fragments are of appropriate size for library preparation.

## Library preparation and sequencing

The NEBNExt Ultra II kit was used for library preparation of all samples in this work. During library preparation, the fragmented DNA was size selected using AMPure beads with a double-sided size selection with bead volumes of 0.5x and 1.3x corresponding to a selected range of about 60 to 500 bp. Library Preparation was done according to the illumina protocol. Sequencing was performed on a NextSeq 2000 with 41-bp paired-end reads.

## DMS cut site analysis

After demultiplexing, the reads were checked for quality using FastQC v0.11.8. Illumina adapters were trimmed using trimmomatic v0.39 (Bolger et al, 2014) (parameters: "ILLUMINACLIP:TruSeq3-PE.fa:2:30:10 SLIDINGWINDOW:10:25 TRAILING:3 LEADING:3 MINLEN:35"). Trimmed reads were aligned to the respective adenovirus genome using bowtie2 v2.4.1 (Langmead and Salzberg, 2012) with parameters --very-sensitive-local --no-discordant. Resulting alignments were filtered for properly paired reads and having a minimum alignment score of 30 with samtools v1.10 (Li et al, 2009). Viral samples yielded alignment rates of 93 to 99%. The relatively low alignment rates of 3.28 and 2.59% for the free DNA replicates can be attributed to contaminating bacterial DNA from the BAC preparations. These samples still yielded coverage values of around 900x, owing to the sequencing depth coupled with the relatively short size of the adenoviral reference (Table 1).

Insert sizes were inferred using Picard v2.21.8 (Picard toolkit, 2019). Deeptools v3.5.0 (Ramírez et al, 2016) was used throughout this work to gain information on coverage, correlation matrices and PCA analyses of alignment files. Information on the nucleotide frequencies along the fragment cut positions was gathered using Homer v4.11.1 makeTagDirectory with the checkGC flag set (Heinz et al, 2010).

The DMS-mediated cleavage sites (DMS footprint) were obtained using a custom Python script, which iterates through all reads and returns a bed file with the cut site of every fragment. The resulting footprint was converted to wiggle format using bedtools (parameters: "coverage -d") (Quinlan and Hall, 2010) and normalized to the number of aligned reads. The global accessibility plots are created by moving a sliding window of size 200 bp and a step size of 10 bp across the genome and counting the cuts in each window.

The yeast DMS-seq data were downloaded for free DNA (run accession DRR066651) and for chromatin (run accession DRR066648) (Umeyama and Ito, 2017; Data ref: Umeyama and Ito 2017). The fastq files were trimmed with trimmomatic (Bolger et al, 2014) and aligned with bowtie2 to the yeast genome (GCA_000146045.2) with the parameters --very-sensitive --no-discordant --no-mixed, in accordance with the original paper. The resulting alignment file was filtered for read pairs with an alignment score of above 30 with samtools (Li et al, 2009). The DMS footprint was extracted with a custom Python script and converted to wiggle format.

## Spectral density estimation (SDE)

The spectral density estimation for the DMS footprint was calculated by the Welch function from the SciPy python module v1.3.1 (Virtanen et al, 2020) (parameters: window = "boxcar", nfft = 60000, detrend = "linear", scaling = "density"). The resulting line plot shows the mean for both samples and the shaded area indicates the region between the lines of both samples. For the yeast DMS-seq data, the periodogram for each chromosome was calculated, and the line shows the mean of all periodograms together with the 95% confidence interval (shaded). For high-resolution heatmaps, the DMS footprint was divided into 700 bp windows with a step size of 35 bp and a periodogram was calculated for each window. These were calculated using the periodogram function from SciPy with the same parameters as the Welch function. The frequencies were converted into periods (1/frequency), while only retaining periods from 50 bp to 130 bp for plotting the heatmaps. The conversion of the linearly distributed frequencies obtained from the periodogram results in exponentially more PSD data points for lower periods. To make the heatmaps more readable, the PSD data points were assigned to the closest integer period, and the median for every period was taken into consideration for the heatmaps. The seaborn package was used to generate heatmaps (Waskom, 2021). The line plots were created by adding up the PSD over all periods from 50 bp to 130 bp for every window.

The spectral density estimation for the dinucleotide repeat patterns was done in a similar way. The distance histograms of the dinucleotides RR (A/G) and YY (T/C) were calculated (see below) in windows of size 700 bp and a step size of 35 bp. Then a periodogram was calculated for every window (parameters: window = "boxcar", nfft = 200, detrend = "linear", scaling = "density"). The maximum power spectral density was assigned to the nearest period integer. Heatmaps are calculated separately for YY and R dinucleotides, but since YY and RR are complementary dinucleotides, the final heatmap is the sum of the two individual ones. The line plot adds up the PSD values for periods 5 to 7 from the heatmap. The line plot was smoothed using a lowess smoother and the peak positions were inferred using the argrelmax function from the scipy.signal module. Peaks were identified similarly from the Ad5-wt and Ad5-ts1 DMS periodicity line plots. For every peak on the viral samples, the distance to the nearest peak from either the set of dinucleotide peaks or random peaks was recorded.

## Dinucleotide distance histograms

A Python script was employed to find the occurrence of every dinucleotide. Because of the small size of the adenoviral genome, we opted to use dinucleotide combinations (e.g., SS stands for GG, GC, CG, or CC). Iterating through the dincleotides coordinates, the distance to every other occurrence was recorded to a maximum of 150 bp downstream, yielding a distance histogram. As described in Bettecken and Trifonov (Bettecken and Trifonov, 2009), the prevalent 3-bp codon pattern of the resulting histogram was smoothed using the mean of a 3-bp window and the mean of a background window of size 5 bp was subtracted to gather the final smoothed distance histogram.

Sequences for human Adenovirus C5 (AY339865.1), human adenovirus C2 (AC_000007.1), human adenovirus E4 (AY487947.1), bat adenovirus 2 (JN252129.1), variola virus (NC_001611.1), and S. cerevisiae (GCF_000146045.2) were downloaded from NCBI and analysed as above.

## V-plots

Bedtools (Quinlan and Hall, 2010) was used to convert the alignment bam files to bed files. A custom Python script was used to extract the fragment length and the fragment midpoint along the reference genome. The heatmap was generated using seaborn, similar to the heatmaps from the spectral density estimations.

## Site-specific protein mutation rates

Hexon and Fiber protein sequences were extracted from the reference genomes, and aligned using MUSCLE. From the multiple sequence alignment, a "dumb" consensus was created, by choosing the most frequent amino acid at every position, if it had a frequency of at least 30%. Every position in each individual protein sequence was compared to the consensus sequence, and the frequency of differences was retained, here termed "mutation frequency".

## Data availability

Raw DMS-seq reads can be found under the BioProject accession PRJNA1219967. The scripts used for creating the visualizations can be found under https://github.com/ConradinBaumgartl/dms_ad_maturation.

The source data of this paper are collected in the following database record: biostudies:S-SCDT-10_1038-S44319-025-00598-z.

## Peer review information

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

## Acknowledgements

High-throughput sequencing was performed at the core facility "Kompetenzzentrum für Fluoreszente Bioanalytik" of the University of Regensburg. Intramural funds of the University of Regensburg supported the study. LEF was supported by a European Molecular Biology Organization (EMBO) long-term fellowship (ALTF356-2018).

## Author contributions

**Conradin Baumgartl**: Conceptualization; Resources; Software; Formal analysis; Validation; Investigation; Visualization; Methodology. **Simon Holzinger**: Conceptualization; Software; Formal analysis; Validation; Investigation; Visualization; Methodology; Writing—original draft. **Uwe Schwartz**: Conceptualization; Software; Formal analysis; Validation; Investigation; Visualization; Methodology; Writing—original draft; Project administration. **Linda E Franken**: Investigation. **Kay Grünewald**: Supervision; Funding acquisition. **Harald Wodrich**: Conceptualization; Resources; Supervision; Investigation; Methodology; Writing—original draft; Project administration; Writing—review and editing. **Gernot Längst**: Conceptualization; Resources; Supervision; Funding acquisition; Validation; Investigation; Visualization; Writing—original draft; Project administration; Writing—review and editing.

Source data underlying figure panels in this paper may have individual authorship assigned. Where available, figure panel/source data authorship is listed in the following database record: biostudies:S-SCDT-10_1038-S44319-025-00598-z.

## Funding

## Disclosure and competing interests statement

The authors declare no competing interests. Linda E. Franken is currently an employee of F. Hoffmann—La Roche Ltd. The presented work was conducted prior to her employment, and this had no influence on the design, analysis, interpretation or reporting of the results herein.

# Expanded View Figures

**Figure EV1.   Depiction of DMS-seq workflow and controls.**

(**A**) Scheme showing the DMS caused fragmentation of DNA by Putrescine and APE1. (**B**) Agarose gel (1.3%) of viral DNA was treated in the absence or presence of 5% DMS for the indicated time points. (**C**) Viral DNA was fragmented after treatment with 2% DMS for 4 min, and a subsequent cleavage reaction was used for library preparation. (**D**) Aligned fragment size distribution histogram of free DNA, Ad-wt, and Ad-ts1. (**E**) Overview of the pipeline for the processing of DMS-seq data. (**F**) Plot of nucleotide frequencies relative to fragment cleavage positions, as indicated.

▶

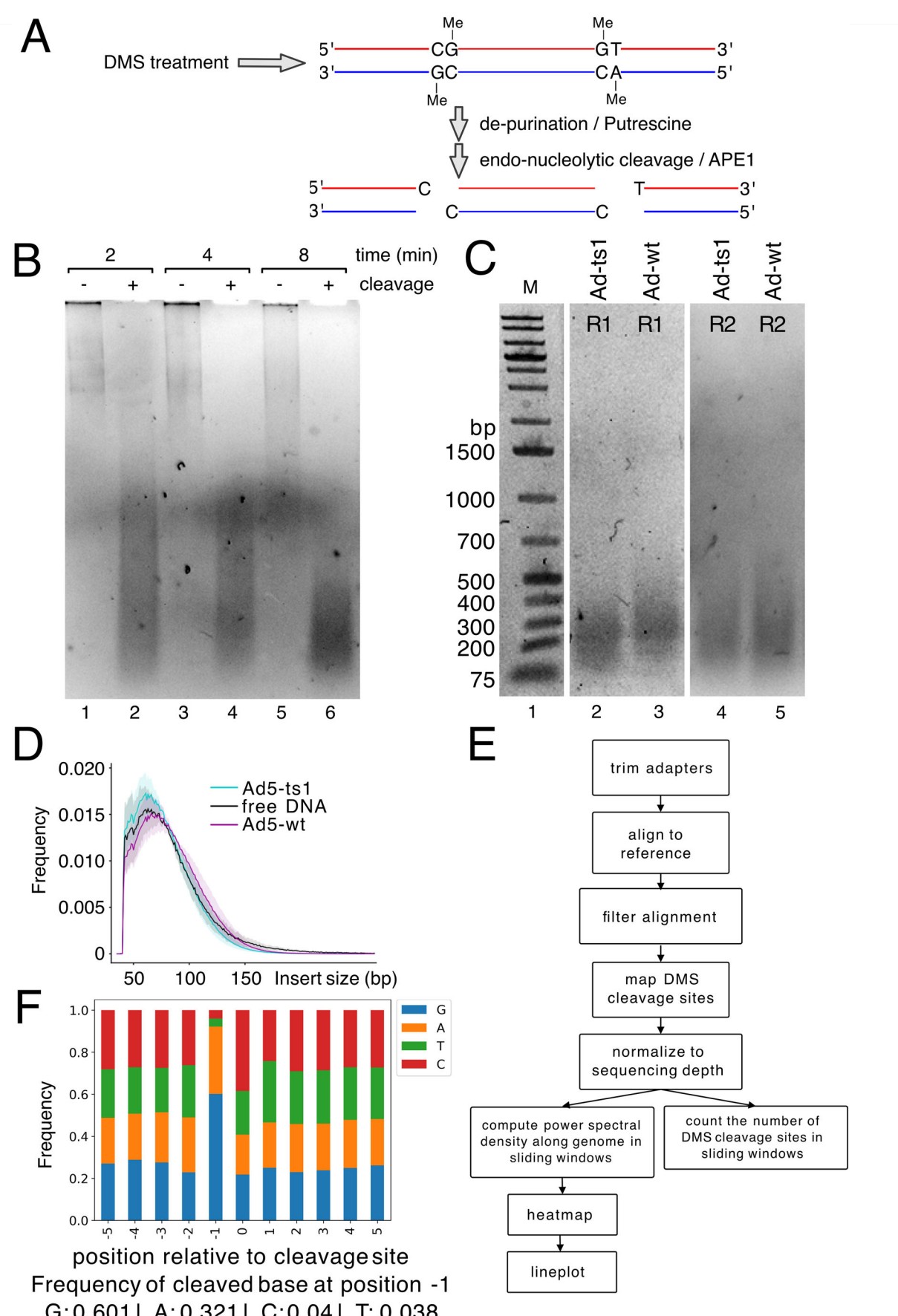

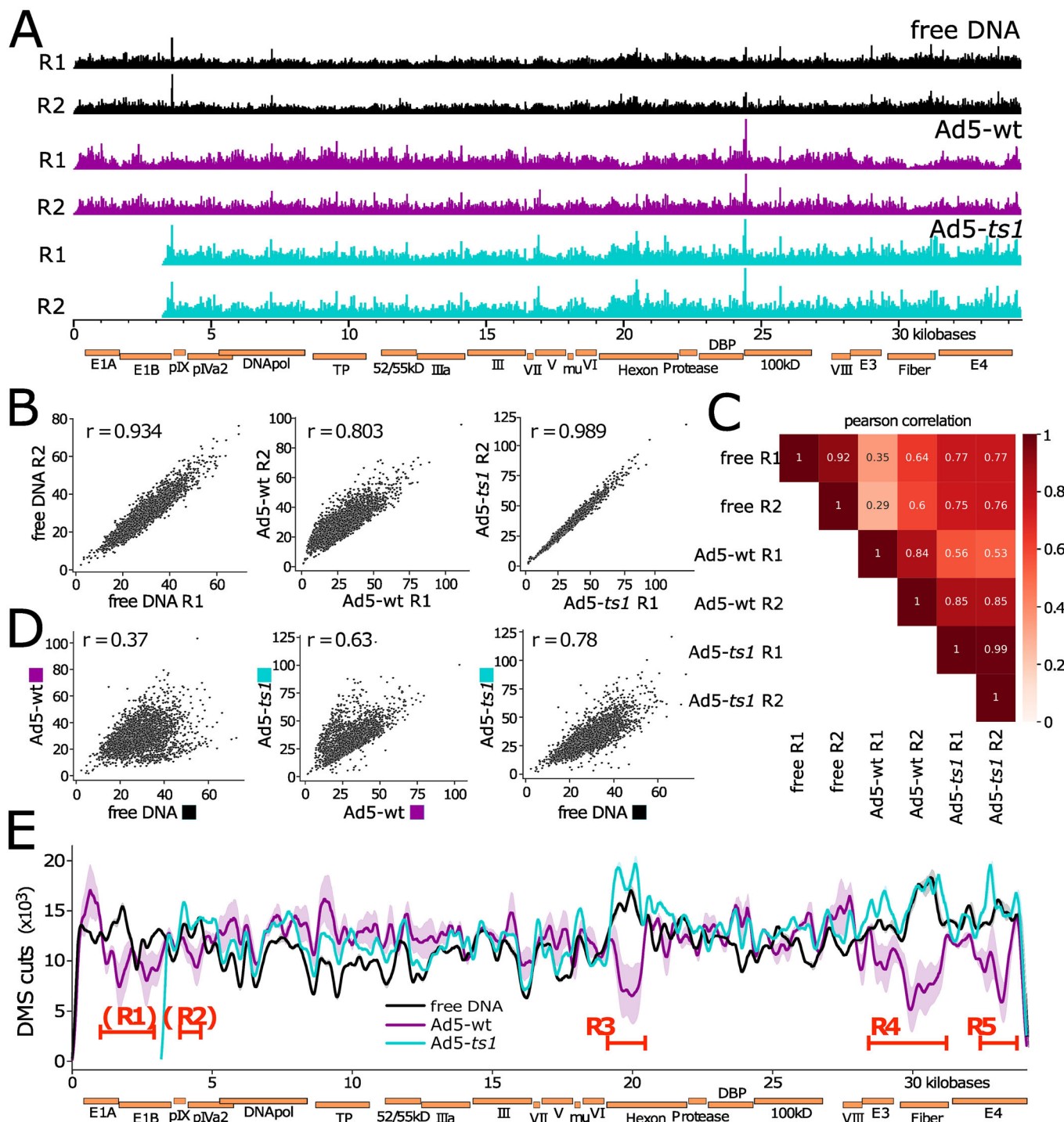

**Figure EV2. DMS cleavage site distribution along the genome.**

(A) Bar chart of cleavage sites along the reference genomes, individually displayed for each replicate. (B) Correlation between replicates in 10 bp bins along the genome. (C) Correlation matrix for the individual samples generated by deeptools ('multiBamSummary bins -bs 5'). (D) Correlation between samples (replicates combined) in 10 bp bins along the genome. (E) Count of DMS cut sites in a 400 bp sliding window with 100 bp step size across the reference genomes.

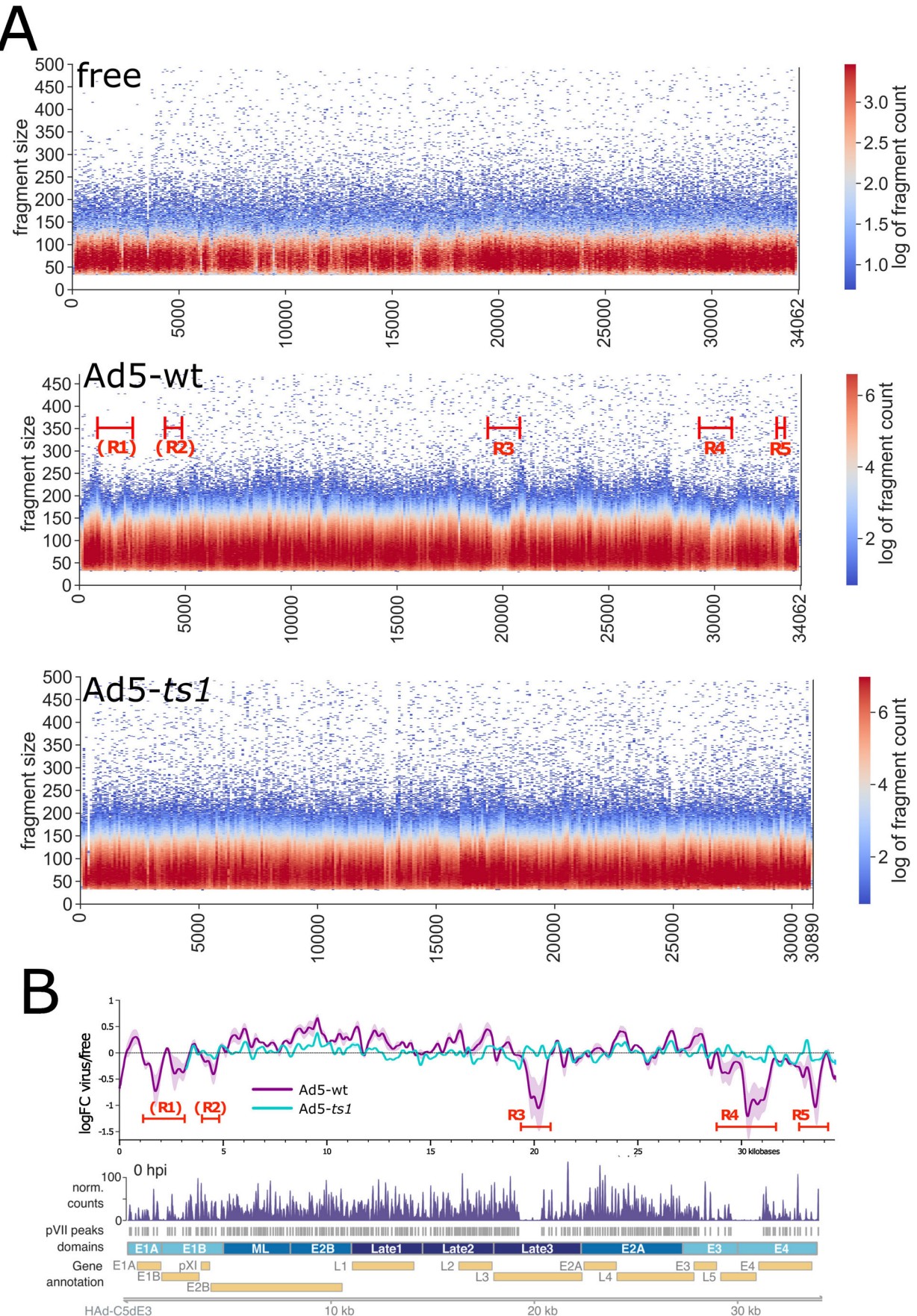

**Figure EV3.   DNA size evaluation along the adenoviral genome.**

(A) V-plots (heatmaps) plotting the fragment size against the fragment midpoint for all samples. (B) Comparison of DMS-seq data with MNase-seq coverage along the adenovirus genome. Top panel: log fold change of DMS methylation sites of viral samples against free DNA (Fig. 1C). Bottom panel: MNase-seq coverage (adapted from Schwartz et al, 2023).

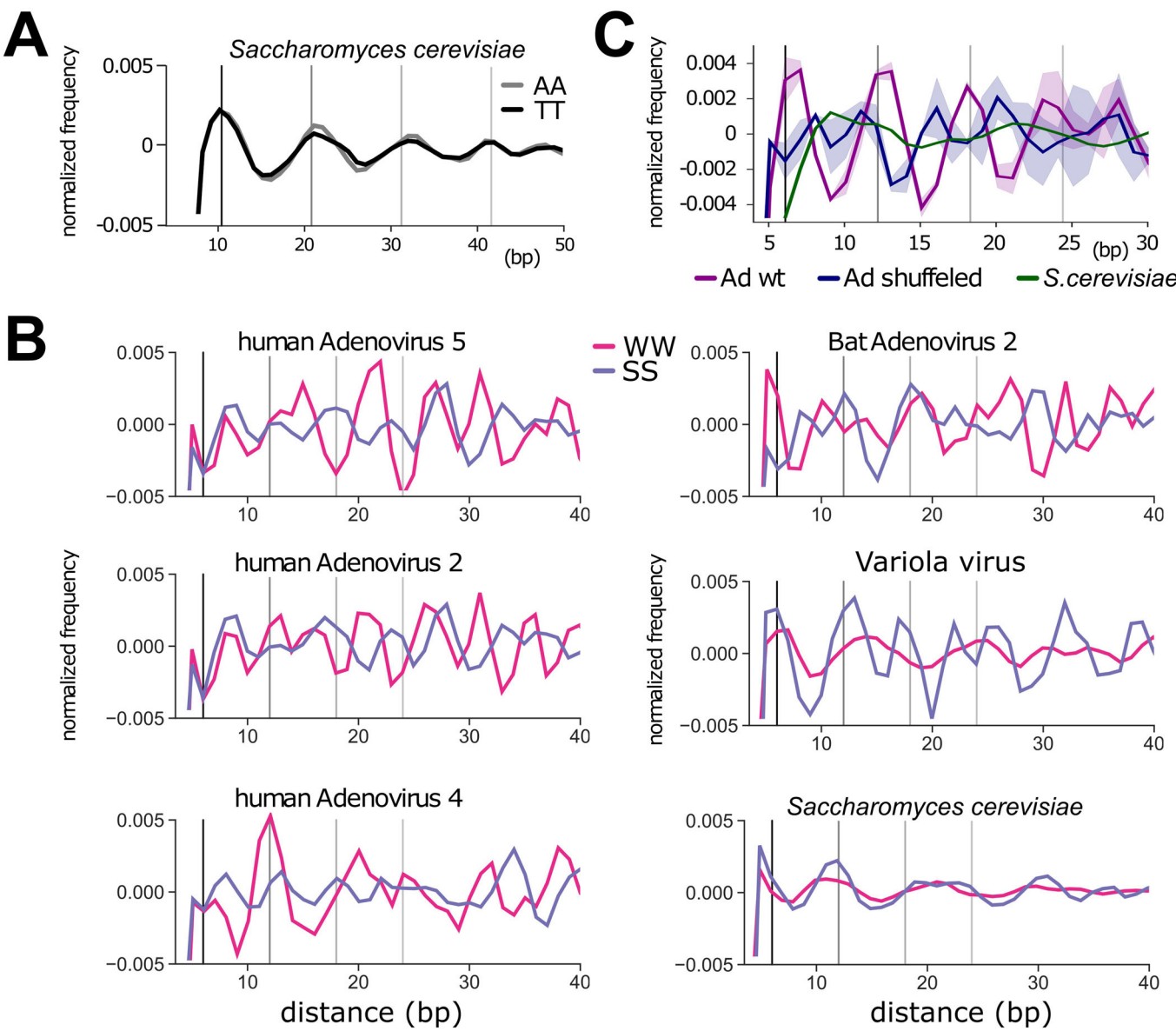

**Figure EV4.  Dinucleotide distance histograms.**

(A) Nucleotide repeat patterns of ~11-bp AA and TT on the S. cerevisiae genome. Vertical lines are placed every 10.3 bp. (B) SS (G/C) and WW (A/T) dinucleotide sequence distance histograms on genomes from different adenovirus species, including Variola virus and the yeast genome. Vertical lines are placed every 6.1 bp. (C) Dinucleotide (~6-bp) periodicity of the T/C (YY) distance histogram on the human adenovirus 5 genome. The same analysis was performed with the shuffled Adenovirus genome and the S. cerevisiae genome. Vertical lines are drawn every 6.1 bp.

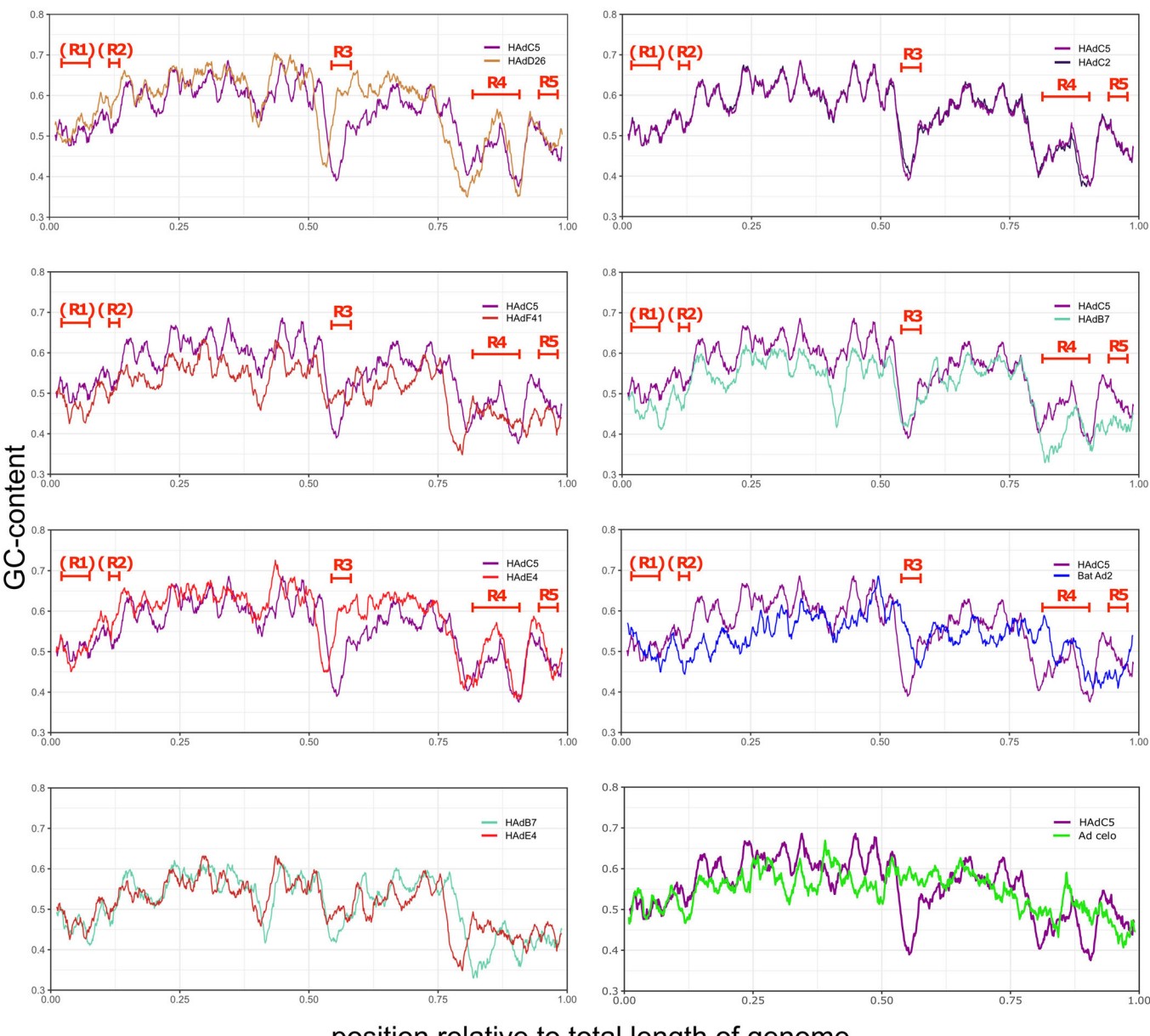

**Figure EV5.  Comparisons of GC-content distribution along various adenoviral genomes.**

The Y-axis shows the GC-content. Because of the variable sizes of the tested genomes, the X-axis represents the relative positions along the entire length of the genome.

