## [Peer Review File · EMBO Reports]

Adenovirus maturation establishes the transcription competent packaging of its genome

Conradin Baumgartl, Simon Holzinger, Uwe Schwartz, Linda Franken, Kay Grünewald, Harald Wodrich, and Gernot Längst

Corresponding author(s): Gernot Längst (gernot.laengst@ur.de), Harald Wodrich (harald.wodrich@u-bordeaux.fr)

Review Timeline:

Transfer Date:	22nd Jul 25
Editorial Decision:	12th Aug 25
Revision Received:	27th Aug 25
Editorial Decision:	19th Sep 25
Revision Received:	24th Sep 25
Accepted:	1st Oct 25

Editor: Achim Breiling / Esther Schnapp

**Transaction Report: This manuscript was transferred to
EMBO reports following peer review at Review Commons.**

**Review
COMMONS**

Review #1

1. Evidence, reproducibility and clarity:

Evidence, reproducibility and clarity (Required)

The manuscript entitled " Adenovirus maturation establishes the transcription competent packaging of its genome" by Baumgartl et al reports interesting results and conclusions on the comparative DMS-sequence analysis of the accessibility of the packaged adenoviral chromosome (adenosome) in the immature and mature virions of Ad5 relative to free viral DNA. The results suggest that while the adenosome is compactly packaged in the immature (ts1) virions, it becomes relaxed upon maturation, thereby increasing the internal pressure as well as transcription competent and priming the mature virions for infection at the right time. However, interestingly the authors identify five regions (R1-R5) that exhibit significantly reduced genome accessibility in mature virions relative to ts1 (Fig. 1C), likely due to DNA reorganization upon maturation.

Looks like the paper was rushed to submission. There are no page numbers nor line numbers included to direct the reviewer's comments.

****Comments and concerns:****

1. A significant comment on the results reported to be addressed is that why does the DMS-seq of free DNA correlates highly with that of ts1 virions (compact state) over the mature virions (relaxed state). I expected the opposite, i.e., the free DNA to be relaxed as there are no histone proteins to condense the free DNA from DMS modifications (methylations) and cuts. Is it possible that free DNA actually contains some (histone-like) basic proteins, picked up from the bacterial cells. Have the authors checked for the presence of proteins in the free DNA samples?
2. Secondly, can authors say anything about how similar or different (i.e., variance of) the adenosome organization from particle to particle would be in either of the particles studied? Is it random or is there some conservation in their organization. This may be addressed by the correlation of reads (by dividing them into two equal groups) at each window (eg., 50-100 bp) along the genome, as in Fig. 1C.
3. The Fig. 1C legend (last sentence) says R1-R5 regions are decreased in the number of DMS cuts in Ad5-wt samples. However, in the Discussion, it is mentioned that in the paragraph beginning with "Using DMS-seq, we identified 5 regions (R1-R5) with increased global DMS accessibility in fully mature Ad5-wt in comparison to immature Ad5-ts1 virions or free viral DNA (Fig. 1). These appear to be contradictory, please clarify.
4. Based on the DMS-methylation, are there regions that exhibit highly similar behavior

between ts1 and wt-Ad5 adenosomes? In other words, the "condensation-state conserved regions". If so, can the authors say anything about those regions?

5. Can authors comment on the changes that could happen to the organization of adenosome and the transcription competency in the case of replication defective or helper dependent (gutless Ads), where a significant adenovirus genome is deleted.

6. Authors say in the last sentence of the paragraph before the section titled "Arrays of phased adenosomes occupy the phased genome", namely " - a feature that has already been established in the virion prior to the cell entry". What is this feature? please specify for the benefit of the readers.

2. Significance:

Significance (Required)

A novel work capturing the organization of adenoviral chromosome, details of which are nearly unknown.

3. How much time do you estimate the authors will need to complete the suggested revisions:

Estimated time to Complete Revisions (Required)

(Decision Recommendation)

Less than 1 month

Yes

Review #2

1. Evidence, reproducibility and clarity:

Evidence, reproducibility and clarity (Required)

****Summary****

In this manuscript, Baumgartl et al. investigate the structural reorganization of the adenovirus genome during viral particle maturation using DMS-seq. They compare wildtype adenovirus (Ad5-wt), a maturation-defective mutant (Ad5-ts1), and protein-free DNA. Their analysis identifies five genomic regions (R1-R5) that become more accessible and are characterized by low GC-content and enriched in dinucleotide periodicity motifs (YY/RR) with a 6.1 bp spacing. These features, the authors argue, are embedded in the DNA sequence and help organize the genome into regular nucleoprotein structures ("adenosomes") during maturation. The study proposes a dual role of core maturation: increasing capsid pressure to facilitate uncoating and pre-organizing the genome for transcriptional activation post-nuclear entry.

The work builds upon a prior study by Schwartz et al. (EMBO J., 2023), which demonstrated sequence-driven positioning of protein VII (pVII) and chromatin changes during early infection using MNase-seq and ChIP-seq. The current manuscript makes a clear step forward by shifting the focus to the in virio state of the adenoviral genome, using a novel DMS-seq-based approach and the ts1 mutant to dissect maturation-dependent chromatin remodeling. It strengthens the view that adenoviral DNA encodes structural information for genome packaging and suggests this organization is primed already prior to nuclear delivery.

****Major Comments****

1. Replication and Robustness

- The authors include two biological replicates per condition (Ad5-wt, Ad5-ts1, and free DNA). Supplementary Figure S2 shows concordance between replicates, supporting a certain robustness of the DMS-seq data at the genome-wide level. However, the main figures presenting differential accessibility (e.g., Fig. 1C) show only smoothed log fold-change values, without displaying replicate-level accessibility profiles across the identified regions (R1-R5). This limits the reader's ability to judge effect size in these critical loci.
- Recommendation: The authors should show replicate signal tracks or coverage plots over regions R1-R5 (and matched control regions) to allow visual assessment of variability and effect magnitude. This would provide a sufficient basis for evaluating robustness without necessitating formal statistical testing.

- Additionally, the authors repeatedly use the term "significant" to describe observed differences, despite having applied no formal statistical tests. This usage is potentially misleading and should be revised throughout the manuscript to avoid confusion. Alternatives such as "notable," "pronounced," or "substantial" would be more appropriate in this context.

2. Inference vs. Demonstration

- The conclusion that specific genomic regions are "primed" for transcription based on increased DMS accessibility is plausible and supported by overlap with MNase data from their earlier work. However, this remains inferential without direct expression data or host factor binding assays.

- Recommendation: Claims about transcriptional readiness should be qualified as hypothesis, unless supported by functional validation.

3. Sequence-Based Chromatin Organization

- The correlation between YY/RR dinucleotide periodicity and phased accessibility is novel and well analyzed using spectral density estimation. While the prior EMBO J. study also suggested a sequence-driven pVII binding code (~5.4 bp WW/SS periodicity near MNase peaks), the current work extends this to a global, genome-wide periodicity (6.1 bp) evident even in free DNA.

- Recommendation: The authors should clarify how the newly identified genome-wide periodicity relates to their prior local motif findings and whether both modes of DNA-based organization are functionally distinct. Claims about functional causality should be softened unless tested via mutational approaches.

4. Mapping efficiency and spectral density of free DNA:

- The mapping rate of reads from the free DNA samples is markedly lower (~2.5-3%) compared to the viral samples (>90%). Do the authors have an explanation for this? Reduced alignment results in lower genome coverage and may contribute to the weaker spectral density signals observed in the free DNA in Fig. 2. While periodicity is still detectable, its amplitude may be dampened due to reduced data density.

- Recommendation: A brief acknowledgment of this limitation would be helpful.

Minor Comments

- Dinucleotide periodicity analysis (Fig. 3A and Supplementary Fig. S4): The term "autocorrelation" is used to describe a custom periodicity analysis based on dinucleotide spacing counts, not statistical correlation coefficients. The y-axis label "normalized

frequency" is undefined and may mislead readers. The authors should clarify the method and rename the y-axis or metric for transparency.

2. Significance:

Significance (Required)

This study provides a conceptual and technical advance in our understanding of adenoviral genome packaging. By shifting the focus to the in virio state using DMS-seq, the authors demonstrate that viral genome accessibility is sequence-programmed and modulated during maturation, even before nuclear entry. This extends prior work by the same group on post-entry chromatin reorganization and transcription initiation.

The insights into maturation-dependent genome decompaction and the identification of evolutionarily conserved, low-GC regions primed for chromatin opening are particularly novel. Moreover, the correlation of global dinucleotide periodicity with phased adenosome structures and their potential implications for vector genome design are conceptually important.

While building directly on earlier work, the new study introduces important new methodology, uses a well-designed comparative mutant system (ts1), and delivers fresh insights into viral genome architecture prior to host interaction.

This work will be of interest to researchers in:

- Functional genomics and epigenomics
- Chromatin biology
- Virology (particularly adenovirus and DNA virus vectors)
- Synthetic biology and vector design

3. How much time do you estimate the authors will need to complete the suggested revisions:

Estimated time to Complete Revisions (Required)

(Decision Recommendation)

Between 1 and 3 months

4. Review Commons values the work of reviewers and encourages them to get credit for their work. Select 'Yes' below to register your reviewing activity at Web of Science

Reviewer Recognition Service (formerly Publons); note that the content of your review will not be visible on Web of Science.

Yes

Review #3

1. Evidence, reproducibility and clarity:

Evidence, reproducibility and clarity (Required)

Conradin Baumgartl and coworkers report studies that explore DNA sequences in the adenovirus genome to determine genome packaging by viral protein VII, which upon core maturation - pVII proteolytic processing - dictate a state of viral chromatin structure that is conducive to viral gene transcription upon entry of the viral genome into the nucleus. The studies exploit a previously established experimental strategy (PMID: 28978481) that was used to detect transcription factors bound to regulatory elements on dsDNA. The strategy relies on dimethyl sulfate (DMS), an alkylating agent that produces N7-methylguanine and N3-methyladenine on exposed regions of double-stranded DNA (dsDNA), followed by depurination and β -elimination which leads to DNA strand cleavage at the methylated sites. Sequencing of the generated DNA fragments was then used to determine the regions in the adenoviral genome that are not protein bound, comparing the adenovirus type 5 (Ad5) wt genome with a maturation deficient virus mutant (Ad5-ts1) that precludes pVII processing. The studies build upon the authors' previous findings which showed pVII protects the viral DNA at defined positions using MNase digestion of the viral DNA, and now reveal that pVII processing during core maturation generates decompacted regions that may be accessible for transcription. The experimental design is clear, and the results in general support the conclusions reached by the authors.

****Major comments/concerns.****

1. The study relies on bioinformatic tools for the comparison of the DNA sequences obtained by DMS-seq from the Ad5-wt virus, the Ad5-ts1 mutant, and free Ad5 DNA, to show that a dinucleotide pattern in the viral genome sequence correlates with variations in the DMS methylation pattern between the three samples, leading to the conclusion that the viral DNA is homogeneously packed in the Ad5-ts1 mutant - in which the pVII cannot be

proteolytically processed - while in the Ad5 wt five conserved regions of the genome with lower GC-content are more DMS sensitive, and hence more protein protected. Although the description of the procedures and data are in general clear, to improve readability and clarity of the bioinformatic methods used I suggest to include a brief description of the parameters chosen in each of the programs/algorithms/python scripts for the DMS cut site analysis, Spectral Density Estimation, dinucleotide autocorrelation and V-plots.

2. The conclusion that nucleotide regions R1 to R5 are decompacted in the Ad5 wt genome, which leads the authors to infer that that "the whole genome enters the nucleus in a "transcription" conducive state in which early transcribed genes are open to become populated with host proteins to drive viral gene expression." should be better justified: i) As described by the authors, since the Ad5-ts1 mutant lacks the E1 region of the viral genome the comparison of the DMS-seq between the Ad5 wt and Ad5-ts1 in this region cannot be made; ii) the R3 and R4 regions correspond to the genes that code for protein IV (fiber) and III (hexon), respectively, which are both expressed during the late phase. Therefore, this conclusion seems to apply only to R2 and R5.

****Other points.****

1. Fig. 1A. Presumably "free DNA" is neither protein-bound, nor associated with viral particles. I suggest to modify the cartoon to indicate this.

2. BioProject accession PRJNA1219967 could not be accessed.

3. To avoid potential ambiguity I suggest to either reconsider or better justify the use of the term "signal" when referring to regions or sequences of DNA.

For example: "The similarity of free DNA and virus periodicity patterns suggests that underlying DNA signals serve as nucleation points for the organization of DNA packaging, with the bound proteins strengthening the periodic pattern (Fig. 2E)."

"However, the spectral density map of Ad5-ts1 is more similar to Ad5-wt than to free DNA (Fig.

2B-D), suggesting that additional signals induce the opening of the regions R3 to R5 (Fig. 1C)."

"We describe here the genomic regions and the signals driving nucleoprotein decompaction, which drive an increase in core pressure required to assist the subsequent capsid disassembly upon entry, for endosomal escape and genome release."

4. Fig. 4. Please describe how "the site specific protein mutation rates between adenoviruses" were determined.

5. Figure S3. Please review the references in the figure caption to Fig. 1C and Fig. 4C.

6. Fig. S1. Panel F: "option"

7. Fig. S1. Panel C: indicate what R1 and R2 stand for; indicate relevant sizes of ladder

bands.

8. Please review writing of the following:

"Every purine base in the one or other strand of DNA serves as a potential target of DMS."

"Putrescine is a diamine that acts as a catalyst for the β -elimination of the remaining sugar in an AP-site and APE1 hydrolyses the phosphor-ester bond of an AP-site at its 5' end."

"Virions were dissolved in 300 μ L water..." Should be resuspended

"...likely as result from the DNA being especially accessible..."

"...theYY/RR repeat pattern for adenosome formation..."

"saccharomyces cerevisiae"

2. Significance:

Significance (Required)

The studies are of relevance as they reveal new insights into the maturation of the virion and structural organization of viral DNA sequences, and opens relevant questions on the evolution and role of viral DNA sequences as templates for nucleoprotein binding that may influence virion stability and infectivity, and determine the pattern of viral gene expression.

3. How much time do you estimate the authors will need to complete the suggested revisions:

Estimated time to Complete Revisions (Required)

(Decision Recommendation)

Less than 1 month

4. Review Commons values the work of reviewers and encourages them to get credit for their work. Select 'Yes' below to register your reviewing activity at Web of Science Reviewer Recognition Service (formerly Publons); note that the content of your review will not be visible on Web of Science.

Yes

Full Revision

Manuscript number: RC-2025-02987

Corresponding author(s): Gernot, Längst

[Please use this template only if the submitted manuscript should be considered by the affiliate journal as a full revision in response to the points raised by the reviewers.]

1. General Statements [optional]

This section is optional. Insert here any general statements you wish to make about the goal of the study or about the reviews.

The reviewer's comments are shown in black color.

The response text by the authors is given in blue color.

Reviewer #1 (Evidence, reproducibility and clarity (Required):

The manuscript entitled " Adenovirus maturation establishes the transcription competent packaging of its genome" by Baumgartl et al reports interesting results and conclusions on the comparative DMS-sequence analysis of the accessibility of the packaged adenoviral chromosome (adenosome) in the immature and mature virions of Ad5 relative to free viral DNA. The results suggest that while the adenosome is compactly packaged in the immature (ts1) virions, it becomes relaxed upon maturation, thereby increasing the internal pressure as well as transcription competent and priming the mature virions for infection at the right time. However, interestingly the authors identify five regions (R1-R5) that exhibit significantly reduced genome accessibility in mature virions relative to ts1 (Fig. 1C), likely due to DNA reorganization upon maturation.

Looks like the paper was rushed to submission. There are no page numbers nor line numbers included to direct the reviewer's comments.

Comments and concerns:

1) A significant comment on the results reported to be addressed is that why does the DMS-seq of free DNA correlates highly with that of ts1 virions (compact state) over the mature virions (relaxed state). I expected the opposite, i.e., the free DNA to be relaxed as there are no histone proteins to condense the free DNA from DMS modifications (methylations) and cuts. Is it possible that free DNA actually contains some (histone-like) basic proteins, picked up from the bacterial cells. Have the authors checked for the presence of proteins in the free DNA samples?

Bacterial Artificial Chromosome (BAC) is a protein-free DNA purified similarly to common plasmids, thus lacking histones or other DNA-associated proteins. DMS, a small molecule, can only produce classic footprints if proteins are quantitatively positioned at the exact same DNA location. For instance, DMS cannot reveal nucleosomal footprints if they are not perfectly positioned on DNA. This is because nucleosomal DNA is methylated by DMS, and only the histone-DNA contact sites are shielded from DMS-dependent methylation. In our ensemble analysis, only quantitative and perfect positioning would lead to DNA methylation inhibition.

Random or weak positioning would yield results similar to those of free DNA. This effect is further enhanced by our method, which employs limited DNA methylation instead of full methylation of all available purines, allowing for substantial DNA fragment lengths necessary for sequencing library preparation.

Given the experimental setup and DMS features (described above), we interpret the similarity between free DNA and *ts1* as a consequence of a more flexible pre-pVII protein positioning, differing in absolute positions between the individual *ts1* virus genomes. The global distribution of cleavage sites is influenced by a minor DNA sequence dependence of DMS-dependent DNA methylation.

2) Secondly, can authors say anything about how similar or different (i.e., variance of) the adenosome organization from particle to particle would be in either of the particles studied? Is it random or is there some conservation in their organization. This may be addressed by the correlation of reads (by dividing them into two equal groups) at each window (eg., 50-100 bp) along the genome, as in Fig. 1C.

In our approach, we are simultaneously gathering information of millions of individual particles, which is often referred to as a composite approach. From this data we can not derive interpretations about individual particles. The ensemble analysis and the partial DNA methylation employed by this method, allows only the statistical analysis of large datasets between two different virus-types in our case, but not within the same viral sample. However, spectral density plots clearly show a defined architecture that must exist in most of the viruses in order to provide statistically significant differences. Also, in our previous study (PMID: 37641864) using MNase digestion of wt-particles shortly after infection, clearly showed a defined pVII occupancy pattern, indicating a similar chromatin organization among the majority of particles.

3) The Fig. 1C legend (last sentence) says R1-R5 regions are decreased in the number of DMS cuts in Ad5-wt samples. However, in the Discussion, it is mentioned that in the paragraph beginning with "Using DMS-seq, we identified 5 regions (R1-R5) with increased global DMS accessibility in fully mature Ad5-wt in comparison to immature Ad5-ts1 virions or free viral DNA (Fig. 1). These appear to be contradictory, please clarify.

As described in the last paragraph of the section titled 'Ad5-wt and Ad5-ts1 exhibit distinct DNA packaging', the lower number of cuts correlates with greater DNA accessibility, as revealed by the V-plots. Compared to the free DNA control, there are fewer DMS cut sites at the R1-R5 regions of the Ad-wt genome. This decrease could be due to increased DNA protection from protein-dependent DNA compaction or, like in our case, could occur from decreased protection of the DNA. The heightened DNA methylation activity of DMS results in very short DNA fragments, which are lost during the DNA sequencing library preparation step. This effect is proven by the V-plots in supplementary figure S3A, which show that regions R1-R5 have lost long DNA fragments (i.e. being more efficiently methylated) and are therefore more accessible to DMS-mediated DNA fragmentation.

4) Based on the DMS-methylation, are there regions that exhibit highly similar behavior between

ts1 and wt-Ad5 adenosomes? In other words, the "condensation-state conserved regions". If so, can the authors say anything about those regions?

Certain regions behave very similarly between Ad5-wt and Ad5-ts1, as illustrated in Figure 2e. The most prominent of these is a peak located around position 23.5 kb of the virus genome. The organisation within this region differs from that of free DNA for Ad-ts1 and Ad-wt, and the structure remains unchanged even following the processing of the wild-type virus. This effect suggests an additional layer of DNA sequence-dependent structural organisation within the virus genome, which operates independently of the GC-content-dependent reorganisation after pre-pVII processing. We propose that further layers of the DNA code or geometric constraints of the particle, beyond di-nucleotide patterns and GC-content, influence the organisation of protein-DNA interactions. However, we did not address this question in this manuscript.

5) Can authors comment on the changes that could happen to the organization of adenosome and the transcription competency in the case of replication defective or helper dependent (gutless Ads), where a significant adenovirus genome is deleted.

We hypothesize that the genome organization is highly dependent on the underlying DNA sequence and/or sequence-dependent structure. Depending on the exact sequence and placement of GC-low regions in the DNA replacing the adenovirus genome, the changes might be beneficial for packaging and/or transduction. Experimental evidence for a sequence dependence of non-coding "stuffer"-DNA was previously shown (PMID: 19515759). The authors showed that inserting prokaryotic stuffer DNA represses transgene expression >60 fold compared to eukaryotic stuffer DNA. Although the authors explain this observation with epigenetic regulation, prokaryotic DNA in this case has a high GC content. In a different study (PMID: 10655474), the authors report similar observations, also showing that the origin of the stuffer DNA greatly affects packaging beyond a simple function in length, again favouring human origin over bacterial DNA. However, these reports remain trial-and-error approaches and are not based on GC content or nucleotide pattern bias.

The potential of stuffer DNA sequence optimisation is indicated in the discussion.

6) Authors say in the last sentence of the paragraph before the section titled "Arrays of phased adenosomes occupy the phased genome", namely " - a feature that has already been established in the virion prior to the cell entry". What is this feature? please specify for the benefit of the readers.

The text was changed for clarity. It now reads: "Indeed, the MNase-seq regions of low compaction do overlap with the regions R3 to R5 from DMS-seq (Fig. S3B). This suggests the DMS data captures the same open chromatin structure of the viral genome - a feature that has already been established in the virion prior to cell entry."

Reviewer #1 (Significance (Required)):

A novel work capturing the organization of adenoviral chromosome, details of which are nearly unknown.

We would like to thank the reviewer for this appreciation and recognizing the novelty of our work.

Reviewer #2 (Evidence, reproducibility and clarity (Required):

Summary

In this manuscript, Baumgartl et al. investigate the structural reorganization of the adenovirus genome during viral particle maturation using DMS-seq. They compare wildtype adenovirus (Ad5-wt), a maturation-defective mutant (Ad5-ts1), and protein-free DNA. Their analysis identifies five genomic regions (R1-R5) that become more accessible and are characterized by low GC-content and enriched in dinucleotide periodicity motifs (YY/RR) with a 6.1 bp spacing. These features, the authors argue, are embedded in the DNA sequence and help organize the genome into regular nucleoprotein structures ("adenosomes") during maturation. The study proposes a dual role of core maturation: increasing capsid pressure to facilitate uncoating and pre-organizing the genome for transcriptional activation post-nuclear entry.

The work builds upon a prior study by Schwartz et al. (EMBO J., 2023), which demonstrated sequence-driven positioning of protein VII (pVII) and chromatin changes during early infection using MNase-seq and ChIP-seq. The current manuscript makes a clear step forward by shifting the focus to the in virio state of the adenoviral genome, using a novel DMS-seq-based approach and the ts1 mutant to dissect maturation-dependent chromatin remodeling. It strengthens the view that adenoviral DNA encodes structural information for genome packaging and suggests this organization is primed already prior to nuclear delivery.

Major Comments

1. Replication and Robustness

- The authors include two biological replicates per condition (Ad5-wt, Ad5-ts1, and free DNA). Supplementary Figure S2 shows concordance between replicates, supporting a certain robustness of the DMS-seq data at the genome-wide level. However, the main figures presenting differential accessibility (e.g., Fig. 1C) show only smoothed log fold-change values, without displaying replicate-level accessibility profiles across the identified regions (R1-R5). This limits the reader's ability to judge effect size in these critical loci.
- Recommendation: The authors should show replicate signal tracks or coverage plots over regions R1-R5 (and matched control regions) to allow visual assessment of variability and effect magnitude. This would provide a sufficient basis for evaluating robustness without necessitating formal statistical testing.

The plots consistently present the complete data from both replicates. The confidence interval for these replicates is indicated by a shaded area within the line plots, particularly noticeable for Ad5-wt in Figures 1C or 2B. The shaded region for Ad-ts1 and DNA is less visible due to the minimal variation in relative signal difference.

Full Revision

- Additionally, the authors repeatedly use the term "significant" to describe observed differences, despite having applied no formal statistical tests. This usage is potentially misleading and should be revised throughout the manuscript to avoid confusion. Alternatives such as "notable," "pronounced," or "substantial" would be more appropriate in this context.

We appreciate the reviewer's feedback and understand the ambiguity associated with the term 'significant'. We have changed the manuscript accordingly.

2. Inference vs. Demonstration

- The conclusion that specific genomic regions are "primed" for transcription based on increased DMS accessibility is plausible and supported by overlap with MNase data from their earlier work. However, this remains inferential without direct expression data or host factor binding assays.
- Recommendation: Claims about transcriptional readiness should be qualified as hypothesis, unless supported by functional validation.

The text passages referring to transcriptional readiness are toned down. We suggest that the open viral chromatin structure established during virus maturation enables nucleosome assembly at the early genes, required for gene activation.

It now reads: "The decompacted regions R1 to R5 are located close to the ends of the viral genome (R1, R2, R4, R5) and more central, overlapping with the hexon gene (R3). The terminal regions being accessible overlap with early genes that are expressed shortly after infection from the incoming genome. This observation suggests that those regions are primed for host protein binding and thus prepared for rapid transcription. "

3. Sequence-Based Chromatin Organization

- The correlation between YY/RR dinucleotide periodicity and phased accessibility is novel and well analyzed using spectral density estimation. While the prior EMBO J. study also suggested a sequence-driven pVII binding code (~5.4 bp WW/SS periodicity near MNase peaks), the current work extends this to a global, genome-wide periodicity (6.1 bp) evident even in free DNA.
- Recommendation: The authors should clarify how the newly identified genome-wide periodicity relates to their prior local motif findings and whether both modes of DNA-based organization are functionally distinct. Claims about functional causality should be softened unless tested via mutational approaches.

The striking similarity between MNase and nucleotide repeat patterns, particularly their periodicity, is noteworthy. The slight variation from 5.4 to 6.1 bp can be attributed to the global analysis (whole virus genome analysis) performed in this manuscript, compared to the rather local analysis (only genomic regions with well-positioned pVII particles). Given that the analyzed genomic regions vary in length and sequence, we observe different values. While MNase and sequence patterns do overlap, we have, as suggested by the reviewer, moderated our text regarding the causal relationship between these patterns.

4. Mapping efficiency and spectral density of free DNA:

- The mapping rate of reads from the free DNA samples is markedly lower (~2.5-3%) compared

Full Revision

to the viral samples (>90%). Do the authors have an explanation for this? Reduced alignment results in lower genome coverage and may contribute to the weaker spectral density signals observed in the free DNA in Fig. 2. While periodicity is still detectable, its amplitude may be dampened due to reduced data density.

- Recommendation: A brief acknowledgment of this limitation would be helpful.

The DNA source is a purified Bacmid with the adenoviral DNA inserted. The presence of bacterial DNA sequences inherent to Bacmid preparations can reduce mapping rates to the target genome. However, our high sequencing depth coupled with the size of the Adenovirus reference enables statistically robust analyses. The Material and Methods section now addresses the differences in DNA origin and mapping rate.

Minor Comments

- Dinucleotide periodicity analysis (Fig. 3A and Supplementary Fig. S4): The term "autocorrelation" is used to describe a custom periodicity analysis based on dinucleotide spacing counts, not statistical correlation coefficients. The y-axis label "normalized frequency" is undefined and may mislead readers. The authors should clarify the method and rename the y-axis or metric for transparency.

The manuscript was edited according to the reviewers' suggestions. The misleading term "autocorrelation" was replaced with "distance histogram".

Reviewer #2 (Significance (Required)):

This study provides a conceptual and technical advance in our understanding of adenoviral genome packaging. By shifting the focus to the in virio state using DMS-seq, the authors demonstrate that viral genome accessibility is sequence-programmed and modulated during maturation, even before nuclear entry. This extends prior work by the same group on post-entry chromatin reorganization and transcription initiation.

The insights into maturation-dependent genome decompaction and the identification of evolutionarily conserved, low-GC regions primed for chromatin opening are particularly novel. Moreover, the correlation of global dinucleotide periodicity with phased adenosome structures and their potential implications for vector genome design are conceptually important.

While building directly on earlier work, the new study introduces important new methodology, uses a well-designed comparative mutant system (ts1), and delivers fresh insights into viral genome architecture prior to host interaction.

This work will be of interest to researchers in:

- Functional genomics and epigenomics
- Chromatin biology
- Virology (particularly adenovirus and DNA virus vectors)
- Synthetic biology and vector design

We appreciate the reviewer's feedback and their recognition of the conceptual advance we aim to convey.

Reviewer #3 (Evidence, reproducibility and clarity (Required)):

Conradin Baumgartl and coworkers report studies that explore DNA sequences in the adenovirus genome to determine genome packaging by viral protein VII, which upon core maturation - pVII proteolytic processing - dictate a state of viral chromatin structure that is conducive to viral gene transcription upon entry of the viral genome into the nucleus. The studies exploit a previously established experimental strategy (PMID: 28978481) that was used to detect transcription factors bound to regulatory elements on dsDNA. The strategy relies on dimethyl sulfate (DMS), an alkylating agent that produces N7-methylguanine and N3-methyladenine on exposed regions of double-stranded DNA (dsDNA), followed by depurination and β -elimination which leads to DNA strand cleavage at the methylated sites. Sequencing of the generated DNA fragments was then used to determine the regions in the adenoviral genome that are not protein bound, comparing the adenovirus type 5 (Ad5) wt genome with a maturation deficient virus mutant (Ad5-ts1) that precludes pVII processing. The studies build upon the authors' previous findings which showed pVII protects the viral DNA at defined positions using MNase digestion of the viral DNA, and now reveal that pVII processing during core maturation generates decompacted regions that may be accessible for transcription. The experimental design is clear, and the results in general support the conclusions reached by the authors.

Major comments/concerns.

1. The study relies on bioinformatic tools for the comparison of the DNA sequences obtained by DMS-seq from the Ad5-wt virus, the Ad5-ts1 mutant, and free Ad5 DNA, to show that a dinucleotide pattern in the viral genome sequence correlates with variations in the DMS methylation pattern between the three samples, leading to the conclusion that the viral DNA is homogeneously packed in the Ad5-ts1 mutant - in which the pVII cannot be proteolytically processed - while in the Ad5 wt five conserved regions of the genome with lower GC-content are more DMS sensitive, and hence more protein protected. Although the description of the procedures and data are in general clear, to improve readability and clarity of the bioinformatic methods used I suggest to include a brief description of the parameters chosen in each of the programs/algorithms/python scripts for the DMS cut site analysis, Spectral Density Estimation, dinucleotide autocorrelation and V-plots.

We included additional information to clarify the procedures. The module version and exact function parameters, as well as the non-default parameters have now been fully described for the Spectral Density analysis. The logic of the employed python scripts, parameters to analyse DMS footprints, as well as the methods for dinucleotide autocorrelation (distance histogram) and the V-plots are now described in more detail in the Materials and Methods section.

2. The conclusion that nucleotide regions R1 to R5 are decompacted in the Ad5 wt genome, which leads the authors to infer that that "the whole genome enters the nucleus in a "transcription" conducive state in which early transcribed genes are open to become populated with host proteins to drive viral gene expression." should be better justified: i) As described by

Full Revision

the authors, since the Ad5-ts1 mutant lacks the E1 region of the viral genome the comparison of the DMS-seq between the Ad5 wt and Ad5-ts1 in this region cannot be made; ii) the R3 and R4 regions correspond to the genes that code for protein IV (fiber) and III (hexon), respectively, which are both expressed during the late phase. Therefore, this conclusion seems to apply only to R2 and R5.

We discuss the potential roles of the regions R1 to R5 in detail, describing their potential function as sites defining early transcription activation and as sites potentially increasing the core pressure by DNA decompaction. In the discussion section, we hypothesize that the different regions have distinct functions.

Other points.

1. Fig. 1A. Presumably "free DNA" is neither protein-bound, nor associated with viral particles. I suggest to modify the cartoon to indicate this.

We have adapted the cartoon in Fig. 1A accordingly.

2. BioProject accession PRJNA1219967 could not be accessed.

The BioProject is now published and can be freely accessed.

3. To avoid potential ambiguity I suggest to either reconsider or better justify the use of the term "signal" when referring to regions or sequences of DNA.

For example: "The similarity of free DNA and virus periodicity patterns suggests that underlying DNA signals serve as nucleation points for the organization of DNA packaging, with the bound proteins strengthening the periodic pattern (Fig. 2E)."

"However, the spectral density map of Ad5-ts1 is more similar to Ad5-wt than to free DNA (Fig. 2B-D), suggesting that additional signals induce the opening of the regions R3 to R5 (Fig. 1C)."

"We describe here the genomic regions and the signals driving nucleoprotein decompaction, which drive an increase in core pressure required to assist the subsequent capsid disassembly upon entry, for endosomal escape and genome release."

We have specified the intended meaning throughout the manuscript, referring directly to "DNA-encoded signal" or "SDE signal". We use the term "DNA-encoded signal" to refer to a DNA sequence/structure encoded feature, similar to the "nucleosome positioning code" in eukaryotic DNA, which is often referred to as a signal (e.g. PMID: 34050142).

4. Fig. 4. Please describe how "the site specific protein mutation rates between adenoviruses" were determined.

A detailed description of the methods is now included in the Materials and Methods section. Briefly, MUSCLE was used to create a multiple sequence alignment (MSA) of the individual protein sequences. The mutation frequency of the DNA sequence relative to the Protein sequence is revealed by comparing the individual protein sequences to the MSA consensus.

5. Figure S3. Please review the references in the figure caption to Fig. 1C and Fig. 4C.

We corrected the legend of Figure S3.

6. Fig. S1. Panel F: "oposition"

The figure text was corrected.

7. Fig. S1. Panel C: indicate what R1 and R2 stand for; indicate relevant sizes of ladder bands.

The information is now given.

8. Please review writing of the following:

"Every purine base in the one or other strand of DNA serves as a potential target of DMS."

The sentence now reads: "Every purine base on either strand of DNA serves as a potential target of DMS."

"Putrescine is a diamine that acts as a catalyst for the β -elimination of the remaining sugar in an AP-site and APE1 hydrolyses the phosphor-ester bond of an AP-site at its 5' end."

The sentence now reads: "Putrescine is a diamine that acts as a catalyst for the β -elimination of the remaining sugar in an AP-site. APE1 hydrolyses the phosphor-ester bond of an AP-site at its 5' end."

"Virions were dissolved in 300 μ L water..." Should be resuspended

The sentence now reads: "Virions were resuspended in 300 μ L water ..."

"...likely as result from the DNA being especially accessible..."

The sentence now reads: "... likely as a result of the DNA being particularly accessible ..."

"...theYY/RR repeat pattern for adenosome formation..."

The missing space was added.

"saccharomyces cerevisiae"

The correct formatting of the species name *Saccharomyces cerevisiae* was applied.

Reviewer #3 (Significance (Required)):

Full Revision

The studies are of relevance as they reveal new insights into the maturation of the virion and structural organization of viral DNA sequences, and opens relevant questions on the evolution and role of viral DNA sequences as templates for nucleoprotein binding that may influence virion stability and infectivity, and determine the pattern of viral gene expression.

Thank you for recognizing the significance of our study highlighting the evolutionary role of DNA sequences beyond their coding capacity.

Dear Prof. Längst

Thank you for the transfer of your revised manuscript from Review Commons to EMBO reports. I have now received the reports from the three referees that I asked to re-evaluate the study, you will find below.

As you will see, the referees fully support the publication of the study in EMBO reports. Referee #2 has a remaining point and suggestion to improve the manuscript I invite you to address in a final revised manuscript. Please also provide a final p-b-p-response with your revised manuscript addressing the remaining concern of referee #1 and the editorial requests below.

The manuscript now needs formatting according to our journal style. Please carefully review the instructions that follow below.

When submitting your final revised manuscript, we will require:

1) a .docx formatted version of the final manuscript text (including legends for main figures, EV figures and tables), but without the figures included. Figure legends should be compiled at the end of the manuscript text.

2) individual production quality figure files as .eps, .tif, .jpg (one file per figure), of main figures and EV figures. Please upload these as separate, individual files upon re-submission.

The Expanded View format, which will be displayed in the main HTML of the paper in a collapsible format, has replaced the Supplementary information. You can submit up to 5 images as Expanded View. I would thus suggest combining the present supplementary figures to have 5 final figure files. Please follow the nomenclature Figure EV1, Figure EV2 etc. The figure legend for these should be included in the main manuscript document file in a section called Expanded View Figure Legends after the main Figure Legends section. Additional Supplementary material should be supplied as a single pdf file labeled Appendix. The Appendix should have page numbers and needs to include a table of content on the first page (with page numbers) and legends for all content. Please follow the nomenclature Appendix Figure Sx, Appendix Table Sx etc. throughout the text, and also label the figures and tables according to this nomenclature.

3) a complete author checklist, which you can download from our author guidelines (<https://www.embopress.org/page/journal/14693178/authorguide>). Please insert page numbers in the checklist to indicate where the requested information can be found in the manuscript. The completed author checklist will also be part of the RPF.

4) that primary datasets produced in this study (e.g. RNA-seq, ChIP-seq, structural and array data) are deposited in an appropriate public database. If no primary datasets have been deposited, please also state this in a dedicated section (e.g. 'No primary datasets have been generated and deposited'), see below.

The accession numbers and database should be listed in a formal "Data Availability" section (placed after Materials & Methods) that follows the model below. This is now mandatory (like the COI statement). Please note that the Data Availability Section is restricted to new primary data that are part of this study. This section is mandatory. As indicated above, if no primary datasets have been deposited, please state this in this section

Data availability

5) We now request the publication of original source data with the aim of making primary data more accessible and transparent to the reader. You will receive a separate email with instructions for providing source data with your revised manuscript, including information how to upload and organize the files.

6) Our journal encourages inclusion of *data citations in the reference list* to directly cite datasets that were re-used and obtained from public databases. Data citations in the article text are distinct from normal bibliographical citations and should directly link to the database records from which the data can be accessed. In the main text, data citations are formatted as follows: "Data ref: Smith et al, 2001" or "Data ref: NCBI Sequence Read Archive PRJNA342805, 2017". In the Reference list, data citations must be labeled with "[DATASET]". A data reference must provide the database name, accession number/identifiers and a resolvable link to the landing page from which the data can be accessed at the end of the reference. Further instructions are available at: <http://www.embopress.org/page/journal/14693178/authorguide#referencesformat>

7) Regarding data quantification and statistics, please make sure that the number "n" for how many independent experiments were performed, their nature (biological versus technical replicates), the bars and error bars (e.g. SEM, SD) and the test used to calculate p-values is indicated in the respective figure legends (also for potential EV and Appendix figures). Please also check that all the p-values are explained in the legend, and that these fit to those shown in the figure. Please provide statistical testing where applicable. Please avoid the phrase 'independent experiment', but clearly state if these were biological or technical replicates. Please also indicate (e.g. with n.s.) if testing was performed, but the differences are not significant. In case n=2, please show the data as separate datapoints without error bars and statistics. See also: <http://www.embopress.org/page/journal/14693178/authorguide#statisticalanalysis>

Please add to each legend (main and EV figures) a 'Data Information' section explaining the statistics used or providing information regarding replicates and scales.

8) Please add scale bars of similar style and thickness to microscopic images, using clearly visible black or white bars (depending on the background). Please place these in the lower right corner of the images themselves. Please do not write on or near the bars in the image but define the size in the respective figure legend.

9) Please also note our reference format:

10) We updated our journal's competing interests policy in January 2022 and request authors to consider both actual and perceived competing interests. Please review the policy <https://www.embopress.org/competing-interests> and update your competing interests if necessary. Please name this section 'Disclosure and Competing Interests Statement' and put it after the Acknowledgements section.

11) We now use CRediT to specify the contributions of each author in the journal submission system. CRediT replaces the author contribution section. Please use the free text box to provide more detailed descriptions and do NOT add an author contributions section to the manuscript text file. See also guide to authors:

<https://www.embopress.org/page/journal/14693178/authorguide#authorshippinguidelines>

12) Please add up to five keywords to the manuscript and order the manuscript sections like this, using these names: Title page - Abstract - Keywords - Introduction - Results - Discussion - Methods - Data availability section - Acknowledgements - Disclosure and Competing Interests Statement - References - Figure legends - Expanded View Figure legends

13) All Materials and Methods need to be described in the main text using our 'Structured Methods' format, which is required for all research articles. According to this format, the Materials and Methods section should include a Reagents and Tools Table (listing key reagents, primers used, experimental models, software, and relevant equipment and including their sources and relevant identifiers), uploaded as separate file, followed by a Methods and Protocols section in which we encourage the authors

to describe their methods using a step-by-step protocol format with bullet points, to facilitate the adoption of the methodologies across labs. More information on how to adhere to this format as well as downloadable templates (.doc) for the Reagents and Tools Table can be found in our author guidelines (section 'Structured Methods'):

14) Please enter all the funding information also into our submission system during resubmission and make sure this is complete and similar to the one mentioned in the acknowledgements section of the manuscript text file.

15) Please remove the list of ORCIDs from the manuscript text. Instead, please supply the ORCID ID for each author in the submission system upon submission of the revised manuscript. Please find instructions on how to link the ORCID ID to the account in our Author guidelines: <http://www.embopress.org/page/journal/14693178/authorguide#authorshipguidelines>

In addition, I would need from you:

- a short, two-sentence summary of the manuscript (not more than 35 words).
- three to four short (!) one sentence bullet points highlighting the key findings of your study.
- a schematic summary figure (synopsis image) in jpeg or tiff format with the exact width of 550 pixels and a height of not more than 400 pixels that can be used as a visual synopsis on our website.

I look forward to seeing a revised version of your manuscript when it is ready.

Please let me know if you have questions or comments regarding the revision.

Best,

Referee #1:

All issues raised in the review have been adequately addressed by the authors.

Referee #2:

I appreciate that most of the concerns raised have been thoroughly addressed.

Regarding the presentation of replicate-level data: I acknowledge the clarification that the shaded confidence intervals in figures represent the variability between replicates. However, this is not clearly indicated in the current figure legends or methods section. To avoid confusion and improve transparency, I recommend that the legends explicitly mention the use of shaded areas to represent replicate variability, and that the method of calculating these confidence intervals is briefly described in the Materials and Methods.

With this minor clarification, all other points have been satisfactorily addressed. I have no further concerns.

Referee #3:

Previous comments/suggestions were adequately addressed. I have no further comments.

Rev_Com_number: RC-2025-02987

New_manu_number: EMBOR-2025-62382V1-T

Corr_author: Längst

Title: Adenovirus maturation establishes the transcription competent packaging of its genome

EMBOR-2025-62382V2 - point-by-point response letter

Dear Editor,

Please find below the point-by-point response to the reviewer comments.

Comments are shown in blue color.

Referee #1:

All issues raised in the review have been adequately addressed by the authors.

Thank you; No further comments.

Referee #2:

I appreciate that most of the concerns raised have been thoroughly addressed.

Regarding the presentation of replicate-level data: I acknowledge the clarification that the shaded confidence intervals in figures represent the variability between replicates. However, this is not clearly indicated in the current figure legends or methods section. To avoid confusion and improve transparency, I recommend that the legends explicitly mention the use of shaded areas to represent replicate variability, and that the method of calculating these confidence intervals is briefly described in the Materials and Methods.

With this minor clarification, all other points have been satisfactorily addressed. I have no further concerns.

The individual legends have been updated as suggested by the reviewer. The changes in the legends are marked in red and now read: "The shaded area represents the range of the two biological replicates and the line shows the averaged value."
Data analysis is described in detail in the Method section, in the chapter "DMS cut site analysis"

Referee #3:

Previous comments/suggestions were adequately addressed. I have no further comments.

Thank you; No further comments.

Dear Gernot,

Thank you for the submission of your revised manuscript and I am sorry for the delay in getting back to you. I have taken over the handling of your study as Achim is currently not in the office. Your ms looks good overall, but a few editorial requests still need to be addressed.

- Your ms has only 4 main figures and should thus be published as a short report with combined Results and Discussion sections. Can you please combine both sections and try to reduce the character count (as much as possible) to close to 30.000 (currently 36449 characters, excluding Methods and References)?
- Please add the email addresses of the corresponding authors on the title page.
- As per journal policy, DATA NOT SHOWN on page 7 is not allowed. Please either add the data or re-phrase.
- The FUNDING INFO is missing both in the ms and in our online submission system. Please add the funding info to both places.
- The Reagents and Tools TABLE needs to be removed from the ms and uploaded as a separate file.
- The Source Data panels for Figure 1 (B and C) are not clearly labeled - each panel should have its own folder

Figure Legends - Comments

- Please define the annotated p values ****/***/**/* as well as provide the exact p-values for the same in the legend of figure 3C as appropriate and reasonable.
- Please indicate the statistical test used for data analysis in the legends of figures 3C
- Please note that the box plots need to be defined in terms of minima, maxima, centre, bounds of box and whiskers, and percentile in the legends of figure 3C
- Please note that information related to n is missing in the legends of figure 3C

I combined the first 2 of your bullet points. Please let me know whether you are OK with this:

- In virio DNA accessibility assay with DMS-seq shows that adenovirus core maturation makes five key viral genomic regions accessible
- These genomic regions are characterized by low GC-content and periodic dinucleotide patterns
- The genomic regions are conserved among adenoviruses

You should be able in our online submission system to bring the old ms files forward to the new ms version and then you can replace only the files that need to be replaced. Please let us know if you have any questions.

Best wishes,
Esther

All editorial and formatting issues were resolved by the authors.

Prof. Gernot Längst
Universität Regensburg
Biochemistry Centre Regensburg
Universitätsstr. 31
Regensburg, Bayern 93053
Germany

Dear Prof. Längst,

I am very pleased to accept your manuscript for publication in the next available issue of EMBO reports. Thank you for your contribution to our journal.

Yours sincerely,
